# Early spatiotemporal dynamics of navigational affordance coding in the dorsal visual cortex

Elisa Zamboni [1,5,7], Rebecca Lowndes[1,7], Richard Aveyard[1], Catriona L. Scrivener[2,6], Jessica A. Teed [2], Yumeng Ma[2], Antony B. Morland [1,3,4] & Edward H. Silson [2] ✉

Successful navigation requires extracting navigationally relevant signals from a dynamically changing visual environment. The process by which we identify navigable routes through the environment is termed navigational affordances. Here, using a combination of functional magnetic resonance imaging, magnetoencephalography and behavioural testing we report that the extraction of such navigational affordance information likely takes place rapidly within dorsal early visual cortex before higher-level scene selective regions. Whilst we replicate prior work showing the involvement of the occipital place area in navigational affordance coding, whole-brain analyses indicate the most likely cortical locus to be dorsal early visual cortex. Analyses comparing the spatiotemporal pattern of navigational affordances suggest such information is detectable within ~110 milliseconds post stimulus onset. Finally, through varying the presentation durations of scenes, we demonstrate that navigational affordance representations are emergent, but not strong with stimulus durations as short as 33-66 milliseconds but become robust with stimulus durations >132 milliseconds. Taken together these data challenge previous views regarding the critical cortical locus for navigational affordance coding and suggest that such affordances can be extracted from very briefly presented stimuli.

Successful navigation through the local environment is a consistent feature of life for mobile organisms[1]. In humans, this accomplishment requires the seamless processing and interpretation of dynamically shifting environments where changes occur rapidly (e.g. exiting the London underground during rush hour).

The human visual system is ideally tuned for such rapid scene processing[2–4], enabling the extraction of relevant visual information to guide navigational and affordance behaviours[5]. Whilst basic-level scene characteristics (e.g. forest or lake) can be deciphered with an exposure of ~50 ms, more global scene characterisations, such as whether a scene is navigable or not, can occur with even briefer presentation times ~34 ms[4]. Cortically, scene processing is thought to rely upon the interactions of a set of scene-selective regions spanning the lateral, ventral and medial cortical surfaces, respectively[6–12]. Within this scene-selective network, prior neuroimaging work of lasting impact suggests the Occipital Place Area (OPA)[7] on the lateral surface plays a

[1]Department of Psychology, University of York, York, UK. [2]Department of Psychology, School of Philosophy, Psychology & Language Sciences, University of Edinburgh, Edinburgh, UK. [3]York Biomedical Research Institute, University of York, York, UK. [4]York Neuroimaging Centre, Department of Psychology, University of York, York, UK. [5]Present address: School of Psychology, University of Nottingham, University Park, Nottingham, UK. [6]Present address: School of Psychology and Neuroscience, University of Glasgow, Glasgow, UK. [7]These authors contributed equally: Elisa Zamboni, Rebecca Lowndes. ✉e-mail: ed.silson@ed.ac.uk

special role in the coding of navigational affordances[13]. The lower visual field bias exhibited by OPA[14–16] is consistent with computational modelling approaches quantifying that the majority of navigational affordance information is extracted from the lower visual field[17].

Within the temporal domain, relatively recent work suggests that neural representations of visual features relevant to navigational affordances within highly simplified artificial environments can emerge as early as ~134 ms post stimulus onset[18]. In contrast, computational work suggests a temporal order of feature extraction, such that the spatial structure of a scene (2D and 3D) is processed first, followed by semantic content and then navigational affordances[19]. Very recent work has extended this temporal framework to show that representations of locomotive action affordances (i.e. what type of action can be performed within a scene), which may be considered analogous to navigational affordances, are present within OPA ~ 200 ms post stimulus onset[20].

Taken together, current models suggest a specific role for OPA in the coding of navigational[13,17] and locomotive action affordances[20,21] that occur within the first few hundred milliseconds post stimulus onset. However, this picture is incomplete. For example, whether OPA should be considered a single functionally homogeneous scene-selective region is debated[15,16,22]. OPA spatially overlaps at least five separate maps of the visual field (Fig. 1A)[15,16], which differ in their visual field representations and thus their potential for extracting navigationally relevant information[13,17]. Moreover, while the spatiotemporal

pattern of responses in OPA has been linked to locomotive action affordances[20,21], this operationalization of affordances differs substantially from previous navigational affordance tasks, which essentially require the identification of efficient routes of egress within scenes[13,18,19]. As such, it is currently unclear what the unique contribution of these maps is to the overall pattern of navigational affordance coding in OPA.

Here, we aim to bridge this gap by leveraging the ability of functional magnetic resonance imaging (fMRI) and Magnetoencephalography (MEG) to capture the brain's spatial and temporal response properties at high-resolution. Participants (n = 14) completed both fMRI and MEG experiments in which they passively viewed scene images taken from prior work on navigational affordances[13]. The spatial and temporal pattern of responses from the fMRI and MEG data were then compared to the navigational affordance representations of these same scenes. An independent group of participants (n = 72) completed a behavioural experiment in which they were required to identify potential routes of egress (i.e. navigational affordances) from the same set of scenes as used in the fMRI and MEG experiments but under different presentation duration conditions (unrestricted, 33 ms, 66 ms, 132 ms, 264 ms and 528 ms). To preview, our data suggests that a likely locus for navigational affordance computations is not OPA, but rather dorsal early visual cortex (V1d/V2d/V3d). Further, navigational affordance representations emerged early in time (~110 ms post stimulus onset) and were relatively transient (~110–450 ms post stimulus

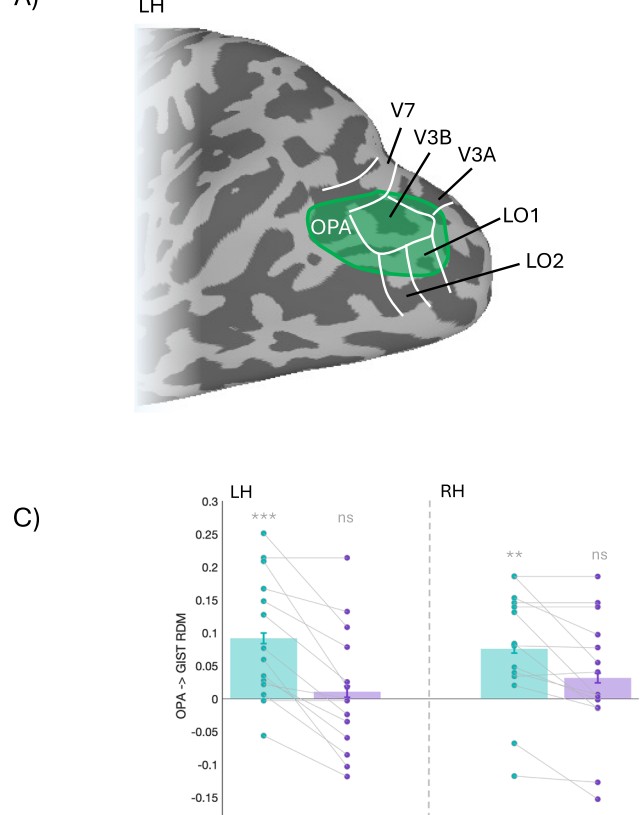

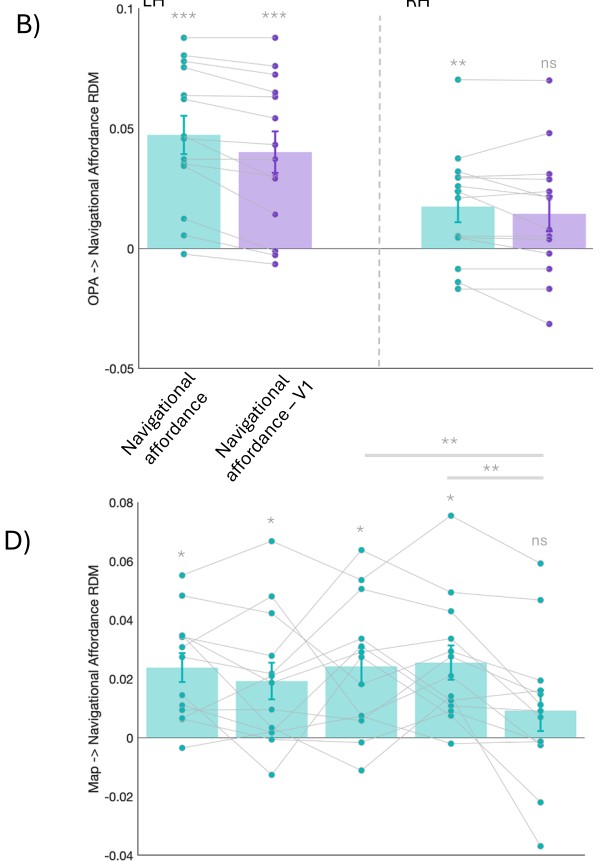

**Fig. 1 | Coding of navigational affordances in OPA and retinotopic subdivisions.** **A** A partially-inflated surface representation of the left hemisphere from a representative participant is shown. The group-averaged OPA ROI (n = 14, Scenes > Faces, p < 0.001, uncorrected) is overlaid in green. The borders of LO1, LO2, V3A, V3B and V7 taken from a probabilistic atlas[28] are overlaid in white. **B** Bars represent the average correlation (n = 14, Spearman's rho) between the OPA RDMs in the left hemisphere (LH) and right hemisphere (RH) and the navigational affordance RDM when considered alone (pale green bars) and when partialling out the V1 RDM (pale purple bars). Values were compared to a no-correlation assumption (i.e. zero) using two-tailed t-tests. **C** Same as (**B**) but for the Gist RDM. **D** Bars represent the average correlation between the RDM of each retinotopic map and the navigational affordance RDM (n = 14, Bonferroni corrected for multiple comparisons). All: Error bars represent the standard error of the mean (SEM). Individual participant data points are shown and linked. *p < 0.05, **p < 0.01, ***p < 0.001, ns= p > 0.05.

onset). This early processing of navigational affordances is consistent with our behavioural data showing that although navigational affordance representations can be extracted with presentation times as brief as 33–66 ms, these representations only become robust with presentation times of 132 ms and longer.

## Results

In this results section, we first present our fMRI findings that are derived from largely replicating a previous study by Bonner et al.[13] This is the necessary platform on which we address our approaches to investigating the role of OPA's retinotopic subdivisions and the timing of navigational affordance coding, which are presented second. Third, through MEG-fMRI fusion[23], representational similarity analysis (RSA)[24] and commonality analysis, we bring together the spatial and temporal dynamics of navigational affordance coding and highlight both the early emergence of navigational affordances and their tendency to be strongest in dorsal early visual cortex. Finally, we ask when meaningful and robust behavioural signatures of navigational affordances emerge, and whether they align with the temporal characteristics of the neural responses we observed.

### Replication of navigational affordance coding in OPA
Our initial analysis goal was to attempt to replicate prior work in which it was shown that the pattern of fMRI responses in OPA to different scene images correlated positively with the pattern of navigational affordance trajectories through those same scenes derived behaviourally[13]. To achieve this, we first computed the pairwise dissimilarity (1-r, where r is the Pearson correlation coefficient) in the fMRI response to all scenes within the left and right OPA, storing these in a representational dissimilarity matrix (RDM)[24]. Next, these participant-specific RDMs were correlated with the behavioural RDM from Bonner et al.[13] to assess the degree of navigational affordance coding in OPA (Fig. 1B). The significance of the correlations with the navigational affordance model was determined via a series of two-tailed t-tests against a zero-correlation assumption. Assumptions were met in all cases.

We observed strong evidence for the coding of navigational affordances in OPA bilaterally (LH: t(13) = 6.14, p < 0.001, Cohen's d = 1.70; RH: t(13) = 2.77, p = 0.015, Cohen's d = 0.77—two-tailed t-tests against zero), replicating prior work. One possibility is that this navigational affordance coding in OPA simply reflects responses to low-level visual features that are inherited from antecedent visual areas like V1. To account for this, we calculated the correlation between the OPA and navigational affordance RDMs in each hemisphere, partialling out the RDM computed from V1. The evidence for navigational affordance coding in OPA remained even after partialling out the V1 RDM in the left hemisphere (t(13) = 4.83, p < 0.001, Cohen's d = 1.33—two-tailed t-test against zero), but not the right hemisphere (t(13) = 2.05, p = 0.060, Cohen's d = 0.57—two-tailed t-tests against zero). Consistent with prior work[13] we also tested whether the OPA RDMs correlated with low-level image statistics derived from the Gist model that describes the orientation and spatial frequency content of images[25]. Importantly, although the OPA RDMs also correlated significantly with one derived from Gist model descriptors[25–27] in both hemispheres (Fig. 1C) (LH: t(13) = 3.64, p = 0.003, Cohen's d = 1.01; RH: t(13) = 3.12, p = 0.008, Cohen's d = 0.86 – two-tailed t-tests against zero), these were found not to survive when considering the contribution of V1 (LH: t(13) = 0.42, p = 0.678, Cohen's d = 0.11; RH: t(13) = 1.22, p = 0.241, Cohen's d = 0.34—two-tailed t-tests against zero). These data replicate evidence for the coding of navigational affordances in OPA, particularly in the left hemisphere[13].

### Navigational affordance coding within retinotopic subdivisions of OPA
OPA spatially overlaps at least five separate retinotopic maps (LO1, LO2, V3A, V3B and V7) to differing degrees[15,16], which raises the question as to whether it should be considered functionally homogeneous. Recently, we showed that the retinotopic subdivisions of OPA do not represent scenes equally, but rather the similarity in their responses is driven largely by the similarity in their visual field representations. That is, maps that are more retinotopically similar (e.g. LO1 and LO2) respond to scenes more similarly[16]. Given this, we next assessed the unique contribution to navigational affordance coding across the retinotopic subdivisions of OPA (Fig. 1D). We were unable to define all five OPA subdivisions in each participant and hemisphere. To account for this, we made use of a probabilistic atlas[28] to define retinotopic ROIs which were constrained to contribute an equal number of nodes (see 'Methods'). Within each map, an RDM was computed based on the pairwise dissimilarity in responses (as above) and these map-specific RDMs were then correlated with the same navigational affordance RDM[13]. Given prior work suggesting that the majority of variance in navigational affordance coding comes from information in the lower visual field[17] we predicted higher correlations in retinotopic maps with a more prominent lower visual field bias (i.e. LO1/LO2) rather than those with either hemifield (V3A/V3B) or more upper visual field representations (i.e. V7). To quantify this effect, the map-specific RSA correlations were submitted to a linear mixed model with Hemisphere (LH, RH) and ROI (LO1, LO2, V3A, V3B and V7) as fixed-effects. Participant was modelled as a random factor. We observed significant main effects of Hemisphere (F(1, 108) = 30.95, p < 0.001) and ROI (F(4, 108) = 3.41, p = 0.010) but no Hemisphere by ROI interaction (F(4, 108) = 1.13, p = 0.345). On average, correlations were higher in the left than right hemispheres (t(108) = 5.56, p < 0.001, Cohen's d = 2.25) and post-hoc comparisons (Bonferroni corrected) revealed only two significant differences with correlations higher in V3A and V3B compared to V7 (V3A versus V7: t(108) = 2.93, p = 0.040, Cohen's d = 0.11; V3B versus V7: t(108) = 3.18, p = 0.019, Cohen's d = 0.08; p > 0.05, in all other cases). We next tested each map-specific RSA correlation against zero (i.e. no correlation assumption) via one-sampled t-tests (Bonferroni corrected). A significant positive correlation was observed for LO1 (t(13) = 4.83, p = 0.002, Cohen's d = 1.39), LO2 (t(13) = 3.08, p = 0.040, Cohen's d = 0.89), V3A (t(13) = 3.86, p = 0.011, Cohen's d = 1.11), V3B (t(13) = 4.36, p = 0.004, Cohen's d = 1.26), but not V7 (t(13) = 1.32, p = 0.210, Cohen's d = 0.38)

### Temporal coding of navigational affordances
We next sought to examine the temporal profile of navigational affordance coding using MEG. Here, RDMs were first constructed from the dissimilarity in the pairwise responses across all MEG sensors at each time-point (see 'Methods') before computing the correlation (Spearman's rho) between (1) the MEG and navigational affordance RDMs and (2) the MEG and navigational affordance RDMs partialling out the contribution of the gist RDM (Fig. 2A, B). Time points with RSA correlations above zero were assessed using Bayesian t-tests[29–32] (for comparison we also include significant time points identified via a frequentist approach on this and all subsequent time-dependent figures, see 'Methods'). We note that overall, the two approaches identify largely equivalent time points (although there are certain cases in which Bayesian evidence for the alternative hypothesis is not identified via a frequentist approach, see 'Methods').

As shown in Fig. 2A, there were at least two distinct time-points with moderate to strong Bayesian evidence (BF$_{10}$ > 3) for a correlation between the MEG and navigational affordance RDMs. An early and relatively transient period peaking ~137 ms post stimulus onset and a later and more sustained period between ~269–381 ms post stimulus onset. Interestingly, the Bayesian evidence for the early transient period weakens when partialling out the contribution of the Gist RDM, but the later more sustained period remains, albeit for a slighter later and narrower time window ~302–389 ms post stimulus onset (Fig. 2B).

Next, we examined the temporal coding of responses within OPA specifically. To achieve this, we employed MEG-fMRI fusion analyses[23].

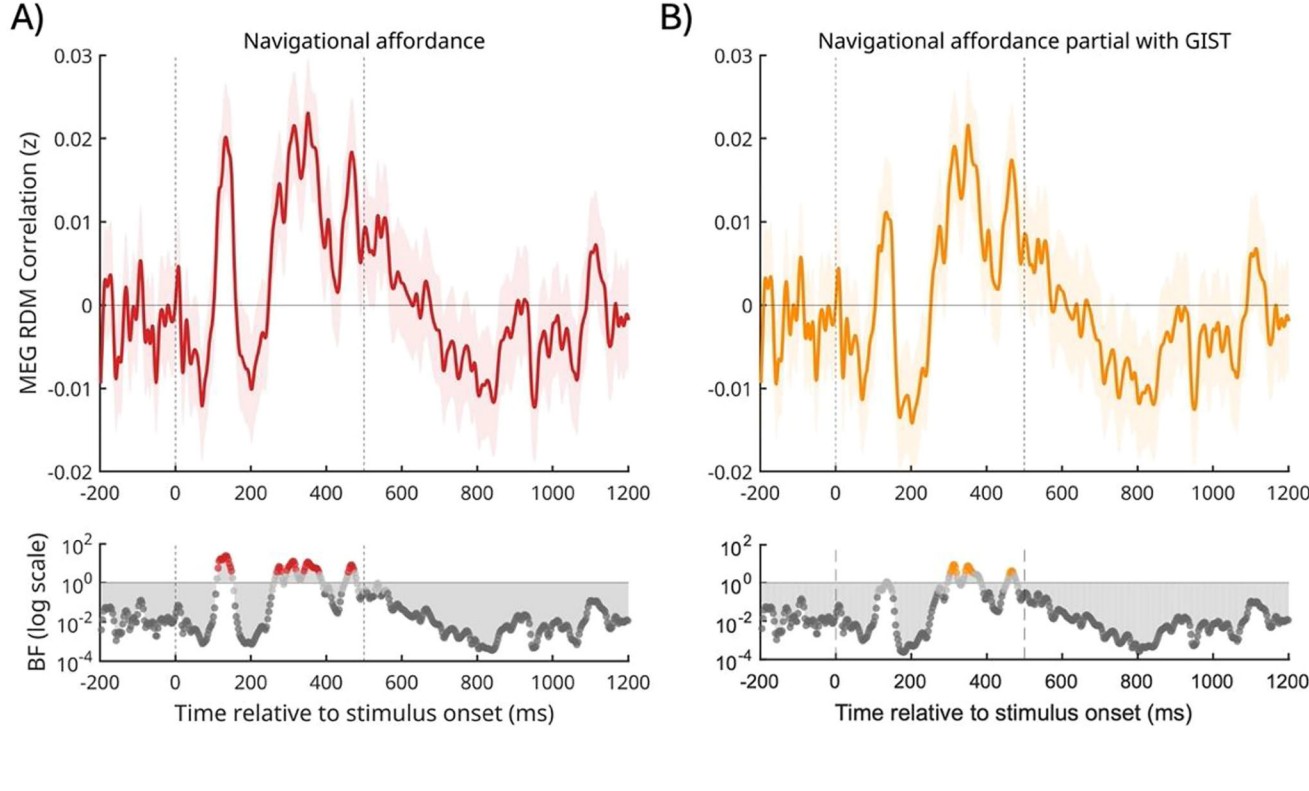

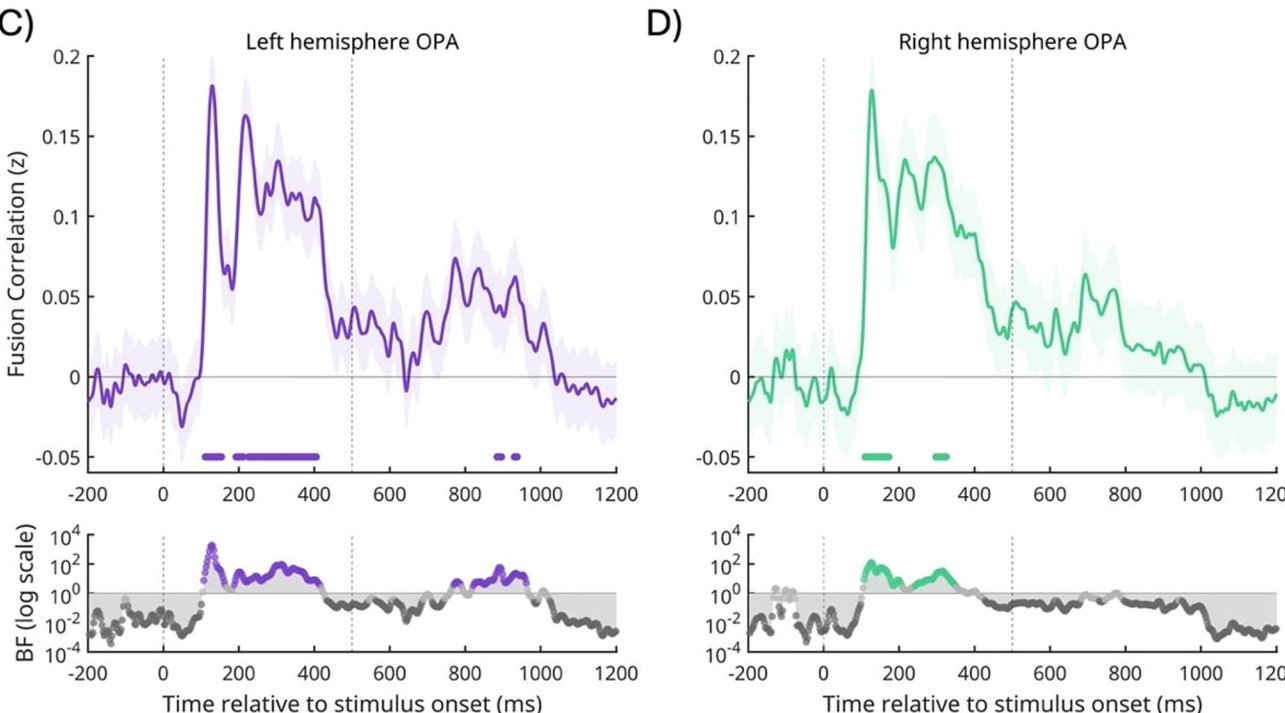

**Fig. 2 | Temporal coding of navigational affordances and MEG-fMRI fusion in OPA. A** The mean RSA timeseries correlation (±SEM) between MEG and navigational affordance RDMs (solid and shaded red). Stimulus onset (time = 0 ms) and stimulus offset (time = 500 ms) are shown by vertical lines. **B** Same as (**A**) but when partialling out the contribution of the Gist RDM (solid and shaded orange). **C** The mean MEG-fMRI fusion timeseries (±SEM) for the left OPA (solid and shaded purple). **D** Same as (**C**) but for the right OPA (solid and shaded green). All: time-points with Bayes Factors that were greater than three are indicated by the coloured circles along the bottom bar, with evidence in favour of the alternative hypothesis (BF$_{10}$ > 3). Dark grey circles indicate Bayes evidence in favour of the null hypothesis (BF$_{10}$ < 1/3). Coloured symbols along the x-axes identify significant time points identified via a frequentist approach (see 'Methods').

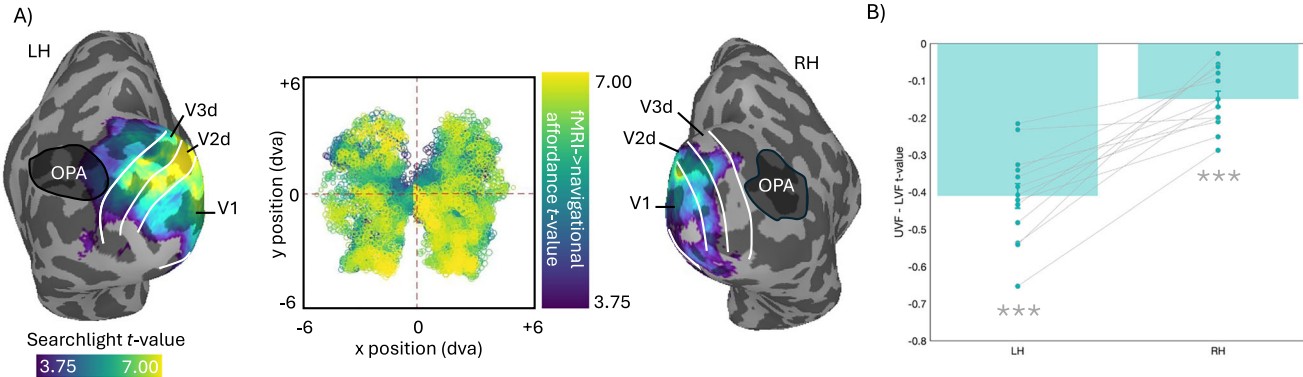

**Fig. 3 | Searchlight and lower visual field biases in navigational affordance coding. A** The strength of the RSA correlation (n = 14, t > 3.75) between fMRI and navigational affordance RDMs partialling out the Gist RDM are overlaid onto partially-inflated surface reconstructions of the left (LH) and right (RH) hemispheres of a representative participant. The group-level OPA ROIs from the Scene localiser are overlaid in black. The borders of V1, V2d and V3d taken from a probabilistic atlas[28] are overlaid in white and provide a good approximation to the location of dorsal early visual cortex. The middle plot represents the group-level RSA correlation t-values as a function of visual field position. **B** Bars represent the mean difference (pale green bars) in the RSA correlation t-values between pRFs with centres in the upper visual field (UVF) minus lower visual field (LVF). Values were compared to a no-bias assumption (i.e. zero) using two-tailed t-tests. In both hemispheres, RSA correlations were significantly larger for pRFs in the LVF (n = 14). Error bars represent the SEM. Individual participant data points are shown and linked. ***p < 0.001.

The aim of this analysis was to identify when in time the representational structure of responses in OPA best matched the representational structure in the MEG time course (see 'Methods'). We calculated the correlation (Spearman's rho) between each participant's OPA RDM (derived from fMRI data) with that participant's MEG RDM at each time point, before averaging across participants (Fig. 2C, D). As above, we assessed time points with fusion correlations above zero using Bayesian t-tests, but also identify significant time points via a frequentist approach for the interested reader. In the left hemisphere, we observed moderate to strong Bayesian evidence (BF$_{10}$ > 3) for MEG-fMRI fusion during two early and relatively transient time-windows, peaking ~118 ms and ~246 ms post-stimulus onset. A later and more sustained period of fMRI-MEG fusion was also present between ~754–980 ms post-stimulus onset (Fig. 2C). In general, evidence for MEG-fMRI fusion was weaker in the right hemisphere, but was nevertheless present for both an early (~113 ms post stimulus onset) and a late (~806–863 ms post-stimulus onset) time window (Fig. 2D).

**Whole-brain searchlight analyses**
Thus far, our analyses have largely considered the spatiotemporal pattern of responses in OPA and its retinotopic subdivisions. To ascertain whether regions outside of OPA also contribute to the coding of navigational affordances we implemented a searchlight approach[33]. For each sphere (6 mm radius around each voxel) we computed the correlation between the fMRI RDM and navigational affordance RDM whilst partialling out the contribution of the Gist RDM (see 'Methods'). Prior work implementing a searchlight approach on fMRI data in response to the same stimuli used here reported evidence for navigational affordance coding in the vicinity of OPA in the right hemisphere, but not the left hemisphere[13].

In our data, whilst we do observe evidence for navigational affordance coding within the left hemisphere OPA, the peak of navigational affordance coding was shown to correspond to dorsal early visual cortex (V1d/V2d/V3d), respectively in both hemispheres (Fig. 3A). Indeed, the spatial extent of significant navigational affordance coding in the right hemisphere approached but did not overlap with our group-based OPA ROI. Importantly, a defining feature of dorsal early visual cortex is an explicit representation of the lower visual field[34], where the vast majority of variance in navigational affordance coding reportedly comes from[17].

We conducted two further analyses to rule out the possibility that this lateralisation was due to systematic leftward biases in path trajectories. First, we have calculated the mean leftward frequency for all 50 scenes in the Bonner and Edinburgh replication datasets. The difference in mean leftward angular frequency across scenes was statistically non-significant (t(49) = 2.00, p = 0.056). This result suggest that unequal sampling of path trajectories is unlikely to explain the differences in lateralisation of the fMRI searchlight data between the two studies. Next, we compared the left versus right trajectories in the Edinburgh replication dataset to rule out a potential trajectory bias in this dataset alone. We calculated how many of the 50 scenes contained trajectories in only one half of the image (i.e. either all left or all right). None of the 50 scenes met this criterion. This indicates that in all 50 scenes path trajectories were in both left and right halves of the image. Given this, we next calculated the mean angular frequency in the left half or right halves of each scene. Such an analysis provides a means to test for a leftward or rightward bias in path trajectories. A paired t-test indicated the difference in mean angular frequency between left and right halves of the scenes to be statistically non-significant (t(49) = 1.10, p = 0.274).

To better visualise the magnitude of navigational affordance coding as a function of visual field position we combined the searchlight analyses with population receptive field (pRF) data (Fig. 3A middle). We first sampled all cortical nodes with significant evidence for navigational affordance coding (t-value > 3.75) and extracted the x, y centre positions of those nodes from the group-average pRF data. The visual field positions of these nodes were then visualised and colour-coded according to the searchlight strength (t-value). In line with prior computational modelling[17], these data indicate that more of the variance in navigational affordance coding is explained by nodes whose pRFs represent the lower rather than upper visual field. To quantify this effect further, we computed an elevation bias in each participant, by taking the average searchlight t-value for pRFs with x, y centre positions in the upper visual field (UVF) minus the lower visual field (LVF), respectively (Fig. 3B). These biases were then tested against zero (no bias assumption) using paired t-tests. In both hemispheres, significantly more of the variance in navigational affordance coding was present in pRFs with centres in the LVF (LH: t(13) = 12.01, p < 0.001, Cohen's d = 3.46; RH: t(13) = 6.71, p < 0.001, Cohen's d = 1.93).

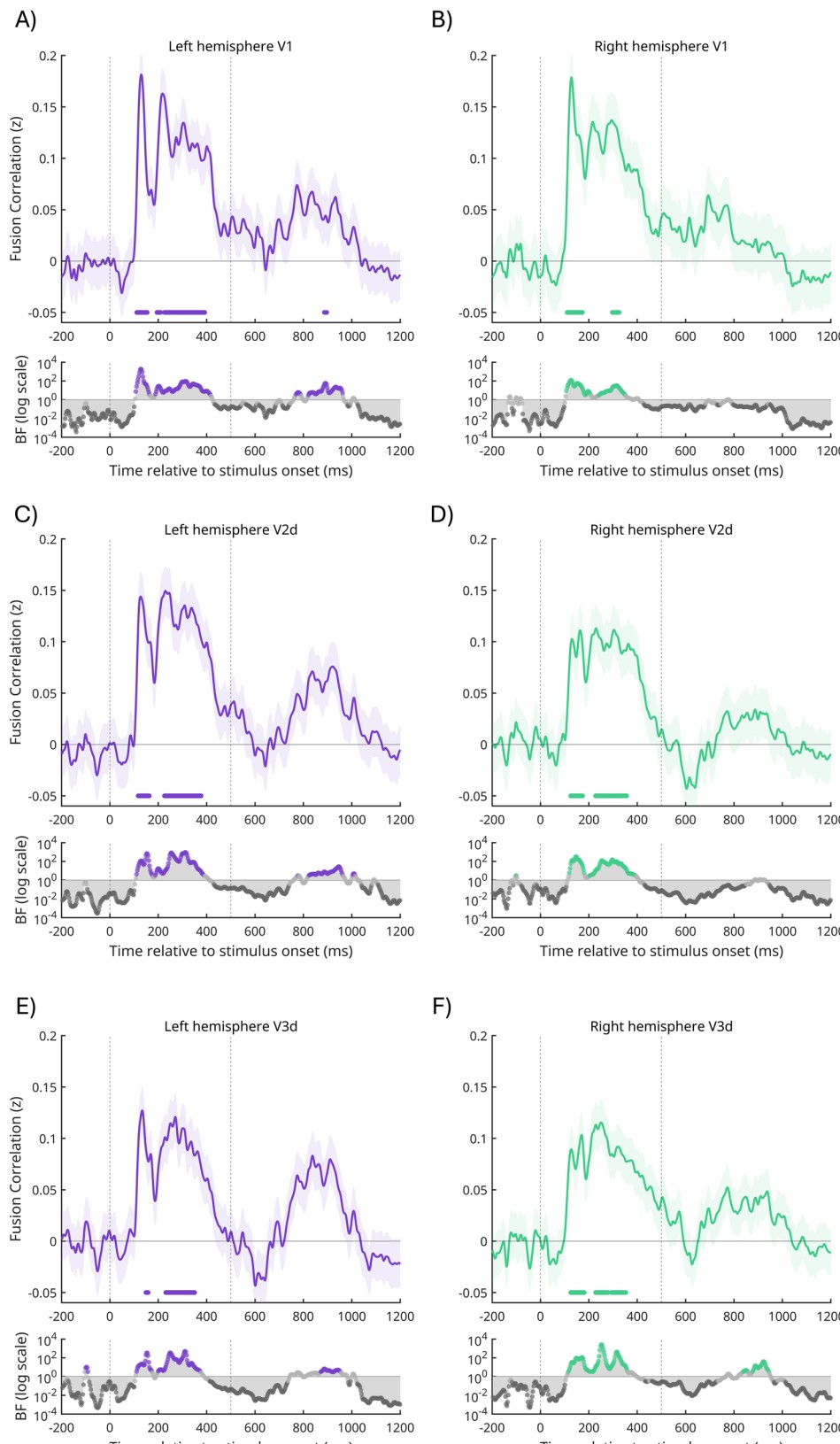

**Fig. 4 | Temporal pattern of MEG-fMRI fusion in dorsal early visual cortex.**
**A** The mean MEG-fMRI fusion timeseries (±SEM) for left V1 (solid and shaded purple). Stimulus onset (time = 0 ms) and stimulus offset (time = 500 ms) are shown by vertical lines. **B**–**F** Same as (**A**) but for right V1 (solid and shaded green), left V2d and V3d (solid and shaded purple) and right V2d and V3d (solid and shaded green), respectively. All: timepoints with Bayes Factors that were greater than three are indicated by the coloured circles along the bottom bar, with evidence in favour of the alternative hypothesis (BF$_{10}$ > 3). Dark grey circles indicate Bayes evidence in favour of the null hypothesis (BF$_{10}$ < 1/3). Coloured symbols along the x-axes identify significant time points identified via a frequentist approach (see 'Methods').

## Temporal pattern of MEG-fMRI fusion for dorsal early visual cortex

The searchlight peaks within dorsal early visual cortex suggests that a likely cortical locus for navigational affordance coding might be earlier in the visual hierarchy than the OPA, despite evidence for OPA's involvement. If responses in dorsal early visual cortex code navigational affordances, then one might predict to see strong and early MEG-fMRI fusion when considering these regions specifically. As a confirmatory step, we applied the same MEG-fMRI fusion analyses above, but now using the fMRI-derived RDMs from V1, V2d and V3d defined using participant-specific pRF mapping data (Fig. 4A–C). It is worth noting that this was not a pre-specified hypothesis, but more a confirmatory check. As above, we primarily assessed the strength of MEG-fMRI fusion using Bayesian t-tests, but also include a frequentist analysis for comparison. In general, a similar pattern of MEG-fMRI fusion was evident across all three ROIs. In V1, we observed moderate to strong Bayesian evidence ($BF_{10} > 3$) for MEG-fMRI fusion during an early and sustained period in the left (-100–480 ms, Fig. 4A) and right hemispheres (-100–450 ms, Fig. 4B). A later period of MEG-fMRI fusion was also present between ~780–980 ms post-stimulus onset in the left hemisphere that was not present in the right. In V2d, we observed moderate to strong Bayesian evidence ($BF_{10} > 3$) for MEG-fMRI fusion during an early and sustained period (-100–400 ms) in both the left (Fig. 4C) and right hemispheres (Fig. 4D). Similar to V1, a later period of MEG-fMRI fusion was also present between ~780–980 ms post-stimulus onset in the left hemisphere that was not present in the right (Fig. 4D). A similar pattern was observed in V3d, with moderate to strong Bayesian evidence for an early and sustained period of MEG-fMRI fusion (-100–400 ms) in both the left (Fig. 4E) and right hemispheres (Fig. 4F). A later period of MEG-fMRI fusion (-880–970 ms) was present in both the left and right hemispheres (Fig. 4F), The MEG-fMRI fusion timecourses are not themselves directly related to the behavioural operationalisation of navigational affordances. To make such a link, we also conducted a commonality analysis between the MEG-fMRI data and the navigational affordance RDM (Fig. 5A, B).

## Temporal coding of navigational affordances in V2d, V3d and OPA

Broadly speaking, the MEG-fMRI fusion data reported above identified two time-windows where the representational structure of the fMRI responses in each ROI best matched the representational structure of the MEG responses; an early time-window (-100–450 ms) and a later time-window (-800–1000 ms). Importantly, the MEG-fMRI fusion analyses do not relate specifically to the representation of navigational affordances. To quantify the unique variance attributable to navigational affordances a commonality analysis was implemented as in ref. 35 (see 'Methods'). Figure 6 depicts the commonality timeseries for V2d, V3d and OPA in both hemispheres, respectively. A similar pattern is evident across hemispheres with unique variance attributable to the navigational affordance model emerging in two stages within the first 500 ms post stimulus onset. An early stage emerging ~75–100 ms, followed by a slightly later stage ~200–400 ms post stimulus onset. Notably, unlike the MEG-fMRI fusion analyses, which indicated both early and late time-windows, the unique variance attributable to the navigational affordances model appears to be restricted to the stimulus presentation window (i.e. 0–500 ms).

## Speed of navigational affordance coding

Taken at face value, the MEG-fMRI fusion and commonality analyses would suggest that the time needed to extract navigationally relevant information from scenes in order to determine efficient routes of egress may, in fact, be surprisingly short. To test this prediction, we ran a behavioural experiment in which we systematically varied the presentation time of each scene across six timing conditions (unrestricted, 33 ms, 66 ms, 132 ms, 264 ms and 528 ms). In each condition,

separate groups of participants (n = 12) viewed each scene before indicating the available navigational routes. These path trajectories were analysed according to the approach described previously[13] (see 'Methods'). In brief, path trajectories were summed across participants before being converted into angular histograms (Fig. 6A). A pairwise dissimilarity RDM of angular histograms was then computed. Initially, we sought to compare the original navigational affordance RDM[13] with our baseline replication RDM (unrestricted condition). A strong positive correlation was observed between the two RDMs (r = 0.63, Fig. 6A). This result provides reassurances that our unrestricted condition captures navigational affordance behaviour in a similar way to that reported previously[13].

Next, RDMs summed across participants were computed for each timing conditon (Fig. 6B). To assess the relationship between conditions we computed the pairwise dissimilarity (1- Pearson's r) between these time-dependent RDMs (Fig. 6C). From the resulting matrix, three general patterns emerged. First, the representation of navigational affordances elicited at 33 ms was very different from all other conditions with the difference increasing as a function of presentation time. Second, the representation of navigational affordances elicited at 66 ms remained different from all other conditions, albeit to a lesser extent than the 33 ms condition. Third, and most crucially, the representations elicited at 132 ms, 264 ms and 528 ms were highly similar to each other. This pattern suggests that robust navigational affordance representations can emerge from presentation times of 132 ms and beyond.

Next, we asked how these time-dependent navigational affordance representations related to both our unrestricted baseline RDM and, for completeness, the original RDM from Bonner et al.[13] To assess the similarity between each time-dependent RDM and our baseline RDM we implemented a bootstrapping analysis. For each timing condition, we randomly shuffled the RDM values before computing the correlation between the baseline RDM and shuffled RDM (repeated 10k times). The resulting distributions represent the relationship between the baseline and time-dependent RDMs one might expect by chance. The observed correlation between the baseline and each time-dependent RDM was then compared against the 95th percentile of the null distributions. The observed correlation fell to the right of the 95th percentile of the null distribution in each timing condition (Fig. 6D). In accord with the relationship between time-dependent conditions mentioned above (Fig. 6C) the correlation with our baseline RDM increased rapidly from weak at 33 ms (r = 0.06), to strong and robust at 132 ms and beyond (132 ms: r = 0.55, 264 ms: r = 0.56; 528 ms: r = 0.56).

For completeness, we also performed the same analysis, but now comparing the observed correlation between the original navigational affordance RDM of Bonner et al.[13] against the time-dependent shuffled distributions (Fig. 6D). Only at the earliest time point (33 ms) did the observed correlation fall to the left of the 95th percentile of the null distribution. In all other conditions, the observed correlation falls to the right of the 95th percentile of the null distribution and becomes strong and robust at 132 ms and beyond. Taken together, we observed strong evidence that the visual information relevant for navigational affordances can be extracted robustly with as little as 132 ms of stimulus presentation. The significant correlations present with briefer stimulus durations (33 ms and 66 ms) should be treated with caution as these navigational affordance representations were shown to be less robust than those elicited at longer durations.

## Discussion

The primary goal of this study was to provide a comprehensive account of the spatiotemporal pattern of navigational affordance coding in human visual cortex. Using fMRI, we found evidence that a likely cortical locus instrumental in navigational affordance coding is dorsal early visual cortex, despite also replicating evidence supporting

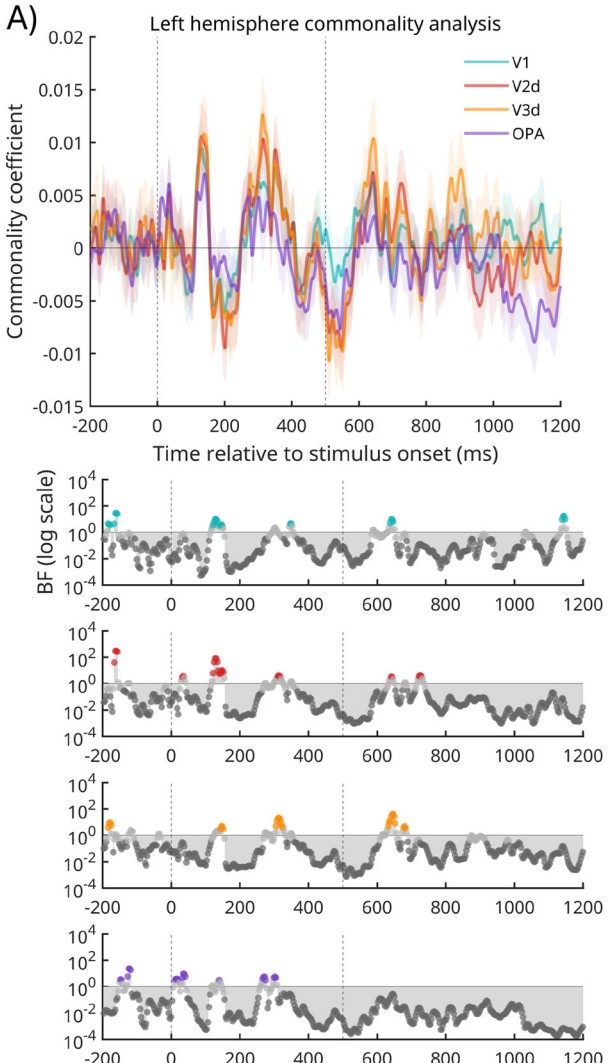

**Fig. 5 | Commonality time-series for V1, V2d, V3d and OPA. A** The mean commonality time-series (±SEM) is plotted for the left hemisphere V1 (green) V2d (red), V3d (orange) and OPA (purple). **B** same as (**A**) but for the right hemisphere. All: timepoints with Bayes Factors that were greater than three are indicated by the coloured circles along the bottom bar, with evidence in favour of the alternative hypothesis ($BF_{10} > 3$). Dark grey circles indicate Bayes evidence in favour of the null hypothesis ($BF_{10} < 1/3$). Coloured symbols along the x-axes identify significant time points identified via a frequentist approach (see 'Methods'). As reported in ref. [35] data are plotted on a quadratic scale. No clusters survived cluster correction via the frequentist approach.

the involvement of OPA in this process. Using MEG and RSA analysis we observed evidence for strong navigational affordance coding within two narrow time-windows, peaking ~110 ms and ~400 ms post stimulus onset and MEG-fMRI fusion analyses within OPA revealed evidence of fusion at both early and late timepoints. Importantly, strong evidence of early MEG-fMRI fusion was also observed in V1, V2d and V3d, respectively. Commonality analyses linking the MEG-fMRI fusion in V2d, V3d and OPA to the representation of navigational affordances directly, suggests that navigational affordances account for unique variance during stimulus presentation only. Finally, we demonstrate that identifying efficient routes of egress can be accomplished with very brief presentation times (33–66 ms), but that these representations only become robust with presentations times of 132 ms and beyond.

Prior work highlighted the human visual systems ability to rapidly extract visual information relevant to complex behaviours at a glance[2–4]. With respect to navigability, it has been shown that stimulus durations of ~34 ms are sufficient for human observers to discern the navigability of scenes very accurately (75% correct)[4]. Our behavioural results are partially inline with this despite the very different way in which affordance behaviour is operationalised between studies. In the

current work, participants were required to identify efficient routes of egress through scenes, which is a qualitatively different behaviour than a binary decision of whether a scene is navigable or not. Although we were able to observe a significant correlation with as little as 33–66 ms presentation, these representations were not strong and only became robust with stimulus durations of 132 ms and beyond. The longer presentation times needed to establish robust representations of navigational affordances, relative to a navigable or non-navigable judgement[4], likely explain the slightly longer durations reported here.

It is not just navigability or navigational affordances that can be gleaned from such brief presentations[3,4]. Indeed, scene categories (e.g. Beach or Forest), naturalness and openness can be deciphered similarly quickly[4]. Feature-level descriptions of scenes briefly presented (~27–500 ms) also tend to precede the reporting of semantic-level information even in the absence of an overt task[36]. It appears therefore that the visual system is capable of extracting visual information valuable to complex cognitive behaviours, such as navigation, from the briefest of glances. Of course, the behavioural goal of the observer likely interacts with the ability to extract goal-relevant visual information[4,37]. More complex goals, such as deciding the correct navigational route to reach a destination, as opposed to simply

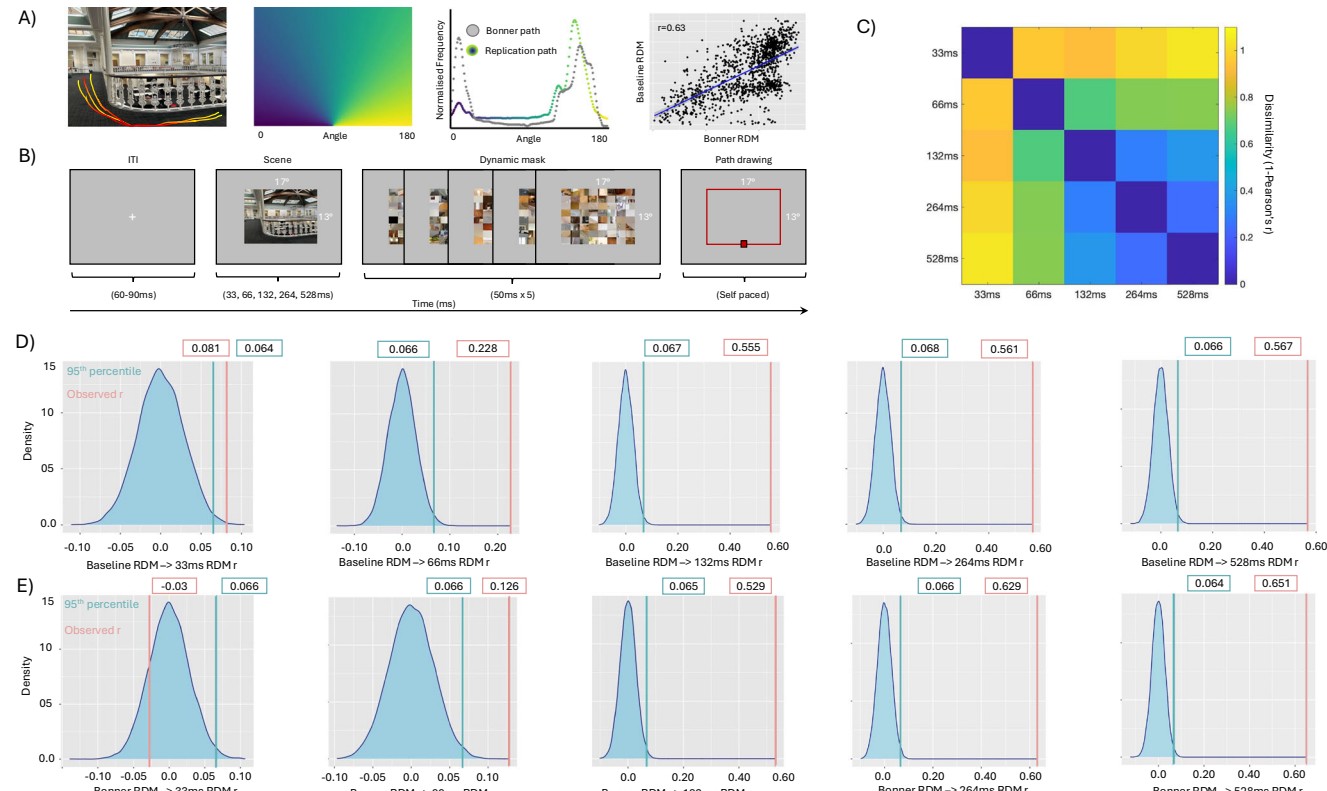

**Fig. 6 | Speed of navigational affordance behaviour. A** Schematic of navigational affordance replication. The image shown in the left-hand panel of (**A**) is for illustrative purposes only and was not shown to participants. Overlaid onto this image are hypothetical path trajectory heatmaps. For each of the 50 scenes used in ref. 13 participant path trajectories were converted into heat maps summed across participants. Heat maps were combined with an angular map to produce an angular histogram per scene. Grey dots represent the angular histogram for the shown scene from ref. 13 with coloured dots representing the angular histogram from our unrestricted baseline condition. A strong positive relationship is present between the navigational affordance RDM from ref. 13 and our replication RDM (r = 0.63). **B** Schematic of the time-dependent task. Following a variable inter-trial-interval, a scene was presented briefly (either 33, 66, 132, 264 or 528 ms) followed by a

dynamic mask (5 × 50 ms). Post mask, the outline of the image remained in red and participants were asked to draw the available routes of egress. The start position of the mouse was fixed on each trial to be the centre and bottom of the scene outline (red square). **C** Time-dependent RDM-RDM matrix. Cells represent the dissimilarity (1-Pearson's r) between all pairs of time-dependent navigational affordance RDMs. RDMs that are more similar are shown in blue with RDMs that are more different in yellow. **D** Distributions represent the correlation (Pearson's r) between the baseline RDM and 10k randomly shuffled RDMs for each of the timing conditions (33, 66, 132, 264 and 528 ms). Green lines represent the correlation at the 95th percentile. Red lines represent the observed correlation between the baseline RDM and each time-dependent RDM. **E** Same as (**D**) but for the correlation between the Bonner RDM[13] and each time-dependent RDM.

identifying the available routes or even simpler, whether a route is navigable or not, will likely increase the exposure time required to reach commensurate performance levels. Very recent work, employing EEG-fMRI fusion suggests that OPA plays an important role in representing locomotive action affordances and that such representations emerge ~200 ms post stimulus onset[20,21]. As mentioned above, the task of identifying the types of movements one can perform within a scene is a different task from identifying efficient routes of egress, although some similarities likely also exist. While the current data suggests that dorsal early visual cortex plays a crucial early role in the latter, it is possible that the more complex task of identifying locomotive affordances relies more heavily on computations within downstream regions, such as the OPA and/or PPA[13,20] and takes advantage of the connections between those regions and other systems.

Our results are both consistent and inconsistent with prior work[13,18–21]. Our fMRI RSA analyses replicated early work showing the involvement of OPA in navigational affordance coding[13]. We extended this work to include an analysis of the retinotopic subdivisions of OPA, with the prediction that the RSA correlations might be higher for retinotopic maps with stronger lower visual field biases, as this is where the majority of navigational affordance information exists[17]. Our hypothesis was not fully supported by the data. Across hemispheres,

the strongest numerical correlations were observed for V3A and V3B, which contain full hemifield representations, as opposed to LO1 and LO2, which in our data were predominantly lower visual field maps. Interestingly, correlations in LO1, LO2, V3A and V3B were all numerically higher than those in V7, which according to our pRF mapping data exhibits the weakest lower visual field bias. This pattern of map specific results should be considered carefully. We were unable to identify each retinotopic subdivision in each individual and so relied upon the use of probabilistic map definitions, which can only be less precise than individually specific map definitions. Further, the fMRI, and MEG-fMRI fusion effects we observed tended to be stronger in the left hemisphere, although very similar patterns were present in the right hemisphere. We urge the interpretation of this difference with caution. It is not immediately clear why a left hemisphere bias should be present in these analyses but could reflect subtle yet consistent differences in temporal signal to noise (tSNR). Additional analyses testing for differences in the path trajectories between the original and replication datasets were statistically non-significant, suggesting a limited effect on the lateralisation we observe.

Our searchlight analysis suggests that the strongest coding of navigational affordances is within dorsal early visual cortex in both hemispheres, although significant evidence for navigational affordances did extend into OPA in the left hemisphere and into ventral

early visual cortex (V1v/V2v/V3v). This contrasts with prior work, which found evidence for navigational affordance coding using the same stimuli in the vicinity of OPA in the right hemisphere only. On the one hand, the localisation of navigational affordances to dorsal early visual cortex is consistent with the premise that the most salient visual features for navigational affordances are derived from the lower visual field[17], which they represent explicitly[34]. One could interpret this result as being due to the contribution of low-level visual features. Importantly, our searchlight analysis is based on the correlation between the fMRI and behavioural RDMs when partialling out the contribution of the Gist RDM, which suggests a low-level visual feature explanation to be unlikely.

In the temporal domain, prior work has suggested that navigational affordance coding occurs relatively late as compared with either the low-level spatial structure or semantic properties of scenes[19]. Our MEG-fMRI fusion analyses in OPA, V2d, V3d and V1 suggest shared representations at both early and later timepoints. On the one hand, the bimodal nature of these temporal responses could be suggestive of an initial and rapid feedforward sweep, followed by a later period of feedback from downstream areas involved in action planning and/or recurrent processing. On the other hand, the commonality analysis, which relates the MEG-fMRI fusion patterns to the navigational affordance model directly, suggests that unique variance attributable to navigational affordances is only present during stimulus presentation. It is important to note that this prior work comparing the EEG responses to behavioural and model RDMs is methodologically different from the MEG-fMRI fusion analyses performed here. Indeed, prior work[18] also comparing EEG and behavioural RDMs found evidence for an early coding of navigational affordances (-134 ms) within a similar time window to that reported here (-110 ms), although it is worth noting that the stimuli used in this prior work were simplified artificial environments and not real-world scenes like those used here.

It is important to distinguish between presentation time and processing time in the context of the current data. Although a significant correlation was present at 33 ms that does not mean that the neural computation occurs as quickly. Indeed, across all the time-resolved analyses reported above, effects emerged -110 ms post stimulus onset at the earliest. As discussed above, the effects reported at 33–66 ms presentation time should be treated with caution. These representations were shown to be very different from all other time-dependent representations. In contrast, the representations of navigational affordances were robust at 132 ms and beyond. It is noteworthy that the duration at which robust representations emerged behaviourally (132 ms) aligns well with the emergence of strong effects in our time-resolved analyses (-110 ms).

If, as our data suggests, navigational affordances can be extracted robustly from brief presentations, what are the implications for the types of visual features that likely drive this process? Prior work highlights the impact that changes in low-level visual features can have on responses within the scene-selective network[26,27,38,39], which are likely to also be reflected by changes in the responses of early visual cortex regions. Our data, combined with computational work indicate that whatever those features are, they are most likely found in the lower visual field as this is where the most navigationally relevant information is located[15,17]. One possibility is that identifying efficient routes of egress requires identifying continuity in the lower visual field. In other words, identifying an absence of obstacles, which in urban environments likely correspond to high spatial frequencies and rectilinear objects[38,40] (e.g. tables, chairs). Future studies should explore the impact of visual feature changes, particularly in the lower visual field, on navigational affordance coding. Finally, it is worth noting that the neuroimaging and behavioural effects reported here and previously[13,17,19] relate only to a small set of relatively homogenous scene stimuli. All of the scenes used here and previously[13,17,19] are indoor urban environments with very clear (but variable) routes of egress. It is currently unclear whether the same pattern of results, particularly the behavioural results, would generalise to a more heterogeneous stimulus set.

In conclusion, by capturing the spatiotemporal dynamics of navigational affordance coding we demonstrate that a likely locus critical to such coding is dorsal early visual cortex. Our characterisation of navigational affordance behaviour also suggests that we can extract visual information relevant to identifying efficient routes of egress through a scene robustly with durations as short as 132 ms.

## Methods
The ethics committee at York Neuroimaging centre at the University of York approved the fMRI and MEG experiments. Ethical approval for the behavioural experiment was provided by the School of Philosophy, Psychology and Language Sciences ethics committee at the University of Edinburgh.

### Participants
14 (10 female) participants with normal or corrected-to-normal vision and a mean age of 24.43 (±4.85) years were recruited for the fMRI and MEG experiments. All participants took part in -2 h of fMRI sessions and 1 h of MEG at the York Neuroimaging Centre. The ethics committee at York Neuroimaging centre at the University of York approved this experiment. 72 (52 female) participants with normal or corrected-to-normal vision and a mean age of 28.51 (±4.10) were recruited for the navigational affordance behavioural experiment. All participants completed the task in one condition only (n = 12 per condition). Ethical approval was provided by the School of Philosophy, Psychology and Language Sciences ethics committee at the University of Edinburgh. All participants provided informed consent and were compensated monetarily for their participation.

### Visual stimuli and tasks
Across all experiments (fMRI, MEG and Behaviour) visual stimuli were presented using PsychoPy. The delivery system used for the visual stimulus in fMRI scans was a ViewPixx projector which projected the stimulus onto a custom-made acrylic screen. The participant viewed the screen with a mirror fixed to the head coil in the scanner. In MEG the stimuli were presented using a ViewPixx Projector which projected the stimulus onto a screen in the MEG suite. Visual stimuli for the behavioural experiment were presented via a HP EliteDesk monitor (resolution: 1920 × 1080 pixels; refresh rate: 60 Hz; visible display size: 531.36 mm × 298.89 mm). We did not receive reports of dropped frames during the experiment. Post data collection we nevertheless tested for dropped frames using the in-built timeByFramesEx.py module within PsychoPy. Out of a sample of 500 frames none were detected as dropped (mean refresh rate = 16.67 ms, standard deviation = 0.03 ms).

### Navigational affordance stimuli fMRI
50 colour images of indoor scenes identical to those used in experiment two of Bonner and Epstein[13]. The navigational affordance of each scene was determined by the previous researchers, who recruited 11 participants to draw the potential routes of egress through each scene. The navigable paths of each scene were summed across participants into a heat map of responses. The 50 scenes used in both the current and the previous study were selected so that the correlations between navigational affordance models and visual models of gist were not reliably greater than chance. For more information on stimuli generation and selection see ref. 13. Each scene consisted of 1024 × 768 pixels and subtended 17 × 13 degrees of visual angle. During fMRI scans, each image was shown for 1.5 s with a 2.5 s inter stimulus interval. Participants took part in 10 runs which each lasted 4 min and 16 s. Each of the 50 scene stimuli were shown in every run. Five catch trials (bathroom images) were also randomly presented in each run

and the participant was instructed to press a button when a bathroom was presented.

## Navigational affordance stimuli MEG

Participants were presented with the same 50 scenes as used in the fMRI navigational affordance experiment. Each scene was presented for 500 ms followed by a 800–1000 ms ISI. Every stimulus was presented eight times in a run, for five runs, leading to 40 presentations per stimulus. Each run lasted 10 min 16 s. 13 participants took part in the experiment as described, one participant took part in ten runs of half the length per run, leading to the same number of presentations per image.

## Behaviour

Participants across all conditions were presented with the same 50 scenes as used in the fMRI and MEG experiments. Following a variable inter-trial-interval (1–1.5 s), a scene was presented briefly (either 33, 66 or 132 ms) followed by a dynamic mask (5 × 50ms). Post mask, the outline of the image remained in red and participants were asked to draw the available routes of egress. The unrestricted condition was similar apart from the lack of a dynamic mask as the scene remained visible throughout the path drawing phase. The start position of the mouse was fixed on each trial to be the centre and bottom of the scene (unrestricted condition) or scene outline (time-dependent conditions). Mask stimuli were created by first dividing each scene (768 × 1024 pixels) into 64 equal sized rectangles (96 × 128 pixels) before shuffling rectangles both within and across scenes.

## Model comparisons

Our initial analyses relied upon the navigational affordance RDM reported for experiment two of Bonner and colleagues[13]. Additionally, we also computed image statistics using the gist descriptor (http://people.csail.mit.edu/torralba/code/spatialenvelope/) to capture the low-level features of each scene. For each scene, a vector composed of 512 values was created by passing each scene through a series of Gabor filters across four spatial frequencies and eight orientations. The resulting vector characterises the image in terms of the spatial frequencies and orientations at different spatial locations within the image. We computed the pairwise Euclidean distance between vectors to produce an RDM across stimuli.

## Scene localiser fMRI

In a block design, participants were presented with images of faces and scenes (different from those used in the navigational affordance experiment), with blocks lasting 16 s. Each block consisted of 20 images presented for 300 ms followed by 500 ms fixation. There were a total of 18 blocks per run. Participants performed a one-back task (button press when same image presented twice in a row). Each run lasted for 5 min and 20 s and each participant completed two runs.

## pRF mapping fMRI

Sweeping bars moved in 8 bar aperture directions which revealed random scene fragments. Each sweep took 36 s to traverse a 17 degree diameter circular aperture in 18 steps. For each step, 5 images were rapidly displayed (2.5 Hz) out of a possible 90 without repeating, so each possible image was displayed once per sweep. Participants performed a colour change detection task at fixation, indicated with a button press. Each run lasted for 5 min 20 s and was repeated four times per participant. This procedure is described in detail in other work[14].

## Data acquisition

**MRI/fMRI.** A single, high resolution, anatomical, T1-weighted scan (TR, 2500 ms; TE, 2.26 ms; TI, 900 ms; voxel size, 1 × 1 × 1 mm³; flip angle, 7; matrix size, 256, 256, 176, total acquisition time, 306 s) was acquired

for each participant. All functional scans consisted of 48 multiband-multiecho EPI interleaved slices (acceleration factor = 2, TEs = 14.6, 32.83, 51.06 ms) within a FOV of 240 × 240 mm with 2.7 mm isotropic voxels (TR = 2000 ms, flip angle = 150°).

**MRI/fMRI preprocessing.** MRI scans were preprocessed using AFNI[41], freesurfer[42] and SUMA[43]. During preprocessing, dummy volumes were first removed from the start of each run (AFNI 3dTcat). Large deviations in signal were removed (3dDespike) followed by slice time correction (3dTshift) aligning each slice with a time offset of zero. The skull was removed from the first echo 1 scan (TE = 14.6 ms) and used to create a brain mask (3dSkullStrip and 3dAutomask), as this echo contained the most signal. The first echo 2 scan (TE = 35.79 ms) was used as a base for motion correction and registration with the T1 structural scan (3dbucket), as this echo was the most similar to standard EPI acquisition. Motion parameters were estimated for the echo 2 scans (3dVolreg) and applied to the other echos (3dAllineate). After completing the standard preprocessing, the data were processed using Tedana[44–47] to denoise the multi-echo scans (version 0.0.12, using default options). Tedana optimally combined and denoised output was then scaled. To do this, we divided the signal in each voxel by its mean value and multiplied the signal by 100 (3dTstat and 3dcalc). This means that the fMRI values can be interpreted as a percentage of the mean signal, and effect estimates can be viewed as percentage change from baseline[48]. For the pRF scans, an average was then calculated across runs to leave a single time series for further analysis. Freesurfer reconstructions were estimated from the T1 anatomical scans (recon-all), and the output used to create surfaces readable in SUMA (SUMA_Make_Spec_FS). The SUMA structural was then aligned to the Session 1 experimental structural to ensure alignment with the functional images (SUMA_AlignToExperiment). Surface-based analyses were conducted using the SUMA standard cortical surface (std.141).

**MEG acquisition and preprocessing.** MEG data was recorded on a 248-channel 4D Neuroimaging Magnes 3600 MEG with electronics upgraded by York Instruments Ltd. Data was acquired at a sampling rate of 1001 Hz with reference channels used to reduce contributions from external noise. Data pre-processing was performed using MNE-Python and consisted of band-pass filtering between 0.05 and 500 Hz, and notch filtering power source noise at 50 Hz and harmonics up to 250 Hz. The time-series data was divided into epochs from 200 ms prior to stimuli onset through to 1200 ms post-onset, and the epochs were demeaned with respect to the 200 ms pre-onset period. Bad channels were automatically detected using the Maxwell filtering utility implemented in MNE-Python, this resulted in a mean exclusion of 5.4 ± 1.2 channels across all recordings.

## MRI analysis

**Regions of interest.** To localise the OPA, a general linear model was estimated for the scene localiser scans using a block design with a 16-s Gamma basis function (GAM: 8.6, 0.547, 16, 3dDeconvolve and 3dREMLfit). The output of the model was then projected onto the SUMA cortical surface (3dVol2Surf), and smoothed with a FWHM of 2 mm (SurfSmooth). OPA ROIs were drawn manually for each subject on the surface (SUMA draw ROI) after thresholding the contrast of scenes versus faces at t > 3.5 (height defining threshold, P < 0.0001, uncorrected). These participant-specific data were applied to a group-level analysis (3dttest++) in order to define group-level ROIs in both hemispheres. Population receptive fields were estimated using AFNI's non-linear fitting algorithm (3dNLfim) and the Gamma (GAM) basis function[14]. The outputs were used to delineate subject-specific V1 ROIs which were drawn manually on the SUMA standard (std.141) surface using the polar angle and eccentricity estimates. Retinotopic maps subdividing OPA (LO1, LO2, V3A, V3B and V7)[15,16] were taken from a probabilistic atlas[28]. To equate ROI size across maps we selected the

top 100 nodes in all ROIs. All ROIs (OPA and retinotopic maps) were converted into 1D files (ROI2dataset) to facilitate future node selection.

**Navigational affordance coding.** The activity associated with each stimulus in the navigational affordance scans were deconvolved using a GAM basis function aligned to the onset of each stimulus. All runs were modelled together, and each stimulus regressor included ten onsets. The data within each participant-specific ROI was then extracted for analysis in MATLAB (ConvertDset) using an RSA framework[24]. The t-values for each stimulus at each node in the ROI were taken from the event-related general linear model (GLM) output. The pairwise dissimilarities between stimulus t-values were then calculated in MATLAB (1−corr, Pearson, v.2021a MathWorks). To quantify the degree of navigational affordance coding in each ROI, we computed the Spearman's correlation (rho) between the navigational affordance RDM and each ROI RDM. To account for the influence of V1 on the observed correlations we employed partial correlation, partialling out the V1 RDM.

Searchlight analyses were performed in Matlab using CoSMoMVPA[33] and the regress_dsms function. For each sphere (6 mm radius around each voxel) we computed the correlation between the fMRI RDM and navigational affordance RDM whilst partialling out the contribution of the Gist RDM. We used centreing of the RDM and converted the correlation values to Fisher's z scores for group level analysis. Volumetric searchlight analyses were submitted to whole-brain voxel-wise t-tests of random effects across subjects before being projected into surface space in SUMA (std.141).

**MEG analysis.** Correlation matrices were generated for each MEG recording run by calculating the Pearson correlation between the pre-processed sensor activity vectors from all pairs of stimuli, at each time sample. Relative dissimilarity matrices were generated by subtracting the resulting correlation values from 1.0, to yield an N-stimuli square matrix for each time sample and each run. The RDMs were averaged across runs by calculating the element-wise mean. This approach was adopted to minimise the impact of head movement between runs.

**MEG-fMRI fusion.** We implemented MEG-fMRI fusion using a representational dissimilarity analysis framework[23]. The aim of this approach is to identify when in time the representational structure across a set of stimuli in a given fMRI ROI is matched in the MEG time course. One benefit of this RSA fusion approach is that the signal is extracted away from the primary MEG and fMRI data formats, which vary considerably in their respective spatial and temporal resolutions[49]. It is worth noting a downside to this approach; if two fMRI regions represent a stimulus set in a very similar way then they will have correlated similarity matrices and therefore similar fusion timeseries, even if this representation emerged at different points in time. However, it is generally accepted that information travelling across regions is transformed non-linearly, meaning that the representational format will likely differ across the brain[23]. To calculate the similarity between the stimulus representation across time and space, we then correlated (Spearman's rho) the MEG RDMs at each time point with the corresponding fMRI RDM for each ROI in each hemisphere. Time points with fusion correlations above zero were assessed using Bayesian t-tests. This was implemented using a half-Cauchy prior with a default width of 0.707 and a range of 0.5 to increase the detection of small effects under the null hypothesis[50].

**Permutation testing of time-resolved data.** Our primary approach for assessing the strength of evidence of time-resolved effects used a Bayesian approach. We note, however, that frequentist approaches are also commonly used and so for completeness we also implemented a permutation-based approach for all time-resolved analyses. Correction for multiple comparisons was implemented in CoSMoMVPA (cosmo motecarlo cluster stat[33]) using threshold-free cluster-enhancement[51–53]. For each analysis we ran 10,000 permutations. For a one-sample t-test and an alpha of 0.05, z-score values greater than 1.65 indicate that the statistic is significant. As we allow for clustering over time, we can make inferences at the cluster level rather than at individual timepoints. A similar approach was adopted for the MEG-RSA analysis taking into account the gist model, for the MEG-fMRI fusion analyses and the commonality analysis. We emphasise here that our approach for assessing the strength of evidence of time-resolved effects used a Bayesian approach. Comparing the results of Bayesian and frequentist approaches can be difficult. For example, it is not clear to us how one should interpret a result if an effect is present in one framework (e.g. Bayesian) but not another (e.g. frequentist). As such, our results are discussed with reference to the Bayesian statistics, but frequentists statistics are also provided for the interested reader.

**Commonality analysis.** We implemented a commonality analysis following the approach described previously[35]. Under this framework, for each ROI, we compared the coefficients of determination ($R^2$) at each timepoint between (1) the group-averaged MEG and fMRI RDMs and (2) the group-averaged MEG and fMRI RDMs when partialing out the navigational affordance RDM. In order to determine significant timepoints we implemented the cluster based analysis described above. For every time-point we randomly shuffled the MEG RDM (1000 times) before calculating the correlation (Spearman's rho). We computed the difference between these two coefficients of determination (for the observed and permuted data) and for each timepoint we assessed whether the observed correlation was greater than the 95th percentile of that timepoints permuted null distribution. We considered windows with more than 3 contiguous timepoints to be significant.

**Behaviour.** Analysis procedure was identical for all conditions and followed the approach outlined previously[13]. Analyses were conducted in Matlab and R (R Studio Version 2024.12.0). The x,y pixel locations of path trajectories for each scene were extracted for each participant. Next, trajectories were spatially blurred (imfilter) before being summed across participants for each scene. Summed heat maps were converted into angular histograms by calculating the summed value in each of 180 angular bins. Angular histograms were then normalised (zscore) and smoothed (smoothn). Finally, RDMs were created by calculating the pairwise dissimilarity in the squared euclidean distance. To create the null distributions for each timing condition, we first randomly shuffled the time-dependent RDM before computing the correlation (Pearson's r) between the shuffled RDM and our replication RDM. This procedure was repeated 10,000 times creating a null distribution of expected correlations by chance. The same procedure was performed to compare the time-dependent RDMs with the original navigational affordance RDM from Bonner and colleagues[13].

### Reporting summary
Further information on research design is available in the Nature Portfolio Reporting Summary linked to this article.

## Data availability
Pre-processed MRI, MEG and behavioural data are available via the open science framework (https://doi.org/10.17605/OSF.IO/PQ2M3). Raw MRI and MEG data will be made available upon request.

## Code availability
Analysis code is available via the open science framework via the open science framework (https://doi.org/10.17605/OSF.IO/PQ2M3).

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

## Acknowledgements

Supported by Biotechnology and Biological Sciences Research Council awards BB/V003887/1 to EHS and BB/V003917/1 to A.B.M., and the School of Philosophy, Psychology and Language Sciences Research Support Grant to J.A.T.

## Author contributions

E.Z., R.L., R.A. collected the fMRI and MEG data. E.Z., R.L., R.A., C.L.S. and E.H.S. analysed the fMRI and MEG data. C.L.S., J.A.T. and E.H.S. designed the behavioural experiment. Y.M. and J.A.T. collected the behavioural data. Y.M., J.A.T. and E.H.S. analysed the behavioural data. A.B.M. and E.H.S. collectively designed the fMRI and MEG experiments. E.H.S. wrote the first draft of the manuscript. All authors contributed to the final version of the manuscript.

## Competing interests

The authors declare no competing interests.
