## [Transparent Peer Review file · Nature Communications]

Early spatiotemporal dynamics of navigational affordance coding in the dorsal visual cortex

Corresponding Author: Dr Edward Silson

Version 0:

Reviewer comments:

Reviewer #1

(Remarks to the Author)

This manuscript uses fMRI/MEG fusion plus a behavioral experiment to examine the neural representation of navigational affordances. The fMRI work replicates the results of Bonner & Epstein (2018), showing OPA involvement in coding navigational affordances. The MEG work shows sustained navigational affordances after 270 ms. The fusion analysis shows V2d/V3d and early (33-66 ms) coding of navigational affordances. A behavioral student presented duration-limited scenes (33.3 ms-132 ms) and asked participants to trace a navigational path. Observers were able to do this with brief presentation times for the most part.

This is a timely paper that examines this problem from multiple perspectives. The involvement of V2d/V3d is novel, and I appreciate the authors' explicit replication of previous work. I am not an expert in the fMRI methods, so I will not comment on this aspect of the paper.

While I feel positively towards this paper, I have two major concerns with the behavioral experiment. The first is logical. As described in the paragraph that starts on line 277, the authors' motivation for this experiment is to verify the rapid coding of navigational affordances revealed by the MEG data. This risks conflating presentation duration with precessing time. Presenting a scene for 33.3 ms and finding that an observer can assess the navigational paths is not the same as the navigational paths being coded in 33.3 ms as the MEG shows. I recommend exampling the paper from Van Rullen (2011) on this fallacy. This interpretation also shows up in lines 309-310 ("Taken together, we observed strong evidence that the visual information relevant for navigational affordances can be extracted in the briefest of glances (~33-66ms)" I would also urge the authors to be more careful in this statement, given the lack of statistical robustness in the earliest presentation time.

My second issue with the behavioral experiment is technical. On lines 568-569 the authors note that the display was "HP EliteDesk monitor (resolution: 1920 × 1080 pixels; refresh rate: 60 Hz; visible display size: 531.36 mm × 298.89 mm)." This means that in the fastest condition, images were on screen for two monitor refreshes. Dropped frames can be very common on PsychoPy (especially on Mac computers, but I have observed it across Linux and Windows machines, too). A dropped frame during a scene presentation would create 50% error and could be a driver of the 33.3 ms condition not showing statistical significance.

As neat as this behavioral paradigm is, the logical and technical issues lead me to believe that this would be a stronger paper without this experiment.

References:

VanRullen, R. (2011). Four Common Conceptual Fallacies in Mapping the Time Course of Recognition. *Frontiers in Psychology*, 2. <https://doi.org/10.3389/fpsyg.2011.00365>

Reviewer #2

(Remarks to the Author)

This paper investigates a concept that has not been extensively studied despite it being very relevant for all motile animals and decades old: affordances. The paper does an in depth study on where affordances are processed in the brain, and the Authors present a clear manuscript with the use of cool approaches and analyses. I have a couple questions/comments (in

order they came up while reading the manuscript):

1. Would you predict that species that may be more tuned into upper visual field (eg: flying/underwater animals, especially eg: killer whales or other marine hunters who find prey above them) may have different affordances? Different representations in the brain?
2. Is action without locomotion the same as affordances?
3. Why left hemisphere effects only? Could this be related to the Figure 5A middle plot of behavioural trajectories which seem to show few turns to the left in your dataset?
4. Why would LO12 effect not be stronger if the argument being made is about lower visual fields' importance in affordance coding?
5. What is the reasoning behind using all MEG sensors?
6. Why not check significant time periods with cluster based stats (which in my opinion are more common for M/EEG analyses) instead of the Bayesian factor?
7. Please include an image of OPA sub-areas tested (LO1, LO2, V3A, V3B & V7) to compare to Figure 3.
8. Figure 5b timing in the image is different from what is in text?
9. How was the accuracy of extraction of path trajectories established?
10. How do the differences in timing in Bonner (2sec) change the comparisons?
11. Because the earliest time of the effect in MEG is ~100ms and not not <66ms, I don't think it's fair to say that the neural effects match the behavioural speed of extraction (and see point 9 above - what does effect at 33ms really mean)?
12. This is just personal but I don't really like the abbreviation "NA" for navigation affordance coding (Figure 1), it's just a little confusing because NA is commonly "not applicable"
13. I am not sure I agree the differences found between Bonner and there are from the fMRI preprocessing, but more likely to be related to performance and/or timing (point 10 too). Also just to be clear, when it is stated that the "behavioral RDM from Bonner et al [11]" you are referring to Fig 2 in the paper.

Reviewer #3

(Remarks to the Author)

Zamboni et al. conducted a study on the coding of navigational affordances in the human brain. For this they combine fMRI, MEG and behavioral data. Their two main claims are that cortical regions V2d/V3d prior to OPA already code for navigational affordances, and that navigational affordances can be extracted from images even with brief glances lasting as little as 66ms. This is an interesting and thought-provoking study, which adds substantially to the literature on navigational coding. It builds a strong foundation by first replicating previous work before moving on to novel analyses.

I have several questions the answers to which should enable me to better determine the strengths of the empirical evidence. I also have suggestions for further analyses and clarifications that may strengthen the manuscript even further.

- 1) I am uncertain what the exact claims of the authors are about the relationship between the two major claims they make (see above). The abstract reads "...we demonstrate that navigational affordances can be extracted within 33-66ms, consistent with navigationally relevant signals emerging early in visual processing.". How is that consistent or inconsistent? A strong interpretation would be that being extractable in a brief glance is a counterpart to signals emerging early in visual processing – but are these not unrelated measures? In the discussion instead those two topics seems to be discussed in a fashion not indicating a close connection. So what is the stance of the authors?
- 2) The results section jumps right in with the first analysis without describing the experimental design and task in any detail. The authors should add a paragraph on what actually happened: what was the paradigm, what was the task, what kind of data was measured to guide the reader.
- 3) It is confusing that some of the region definitions that are possible based on retinotopic mapping are based on participant-specific pRF estimates, and some are not, instead relying on anatomical atlases. Those atlases can only be less precise than participant-specific mapping. This is important when for example differentiating OPA from more posterior regions. For the claims of the manuscript this is not a major downside as arguably, even with better localization, there would be no other regions posterior to OPA than V2/3d. However, the limitation should be acknowledged and possible implications discussed.
- 4) l. 134 and before. The emerging results pattern is complex. What do the authors finally conclude from the results? Please clarify.
- 5) Assessing the temporal coding of navigational affordances the authors make a point differentiating early from late responses. I suggest a temporal generalization analysis to determine the relationship between the representations at earlier versus later time points, across analyses in the manuscript.
- 6) Based on the results reported in Fig. 2D the authors claim that navigational affordances are already coded at the border of

V23d. For one, I am uncertain about whether the expression "at the border" is justified, given that we are dealing with group-average anatomically determined borders. Second, there are also strong effects posterior to V2, in V1, or, using the authors' terms, at the border of V1 and V2d (especially in the right hemisphere. How do the authors interpret this? I also suggest that the authors add a MEG-fMRI fusion analysis for V1.

7) Figure 4 reports strong effects after stimulus offset. How is this to be interpreted?

8) The authors conduct fusion analysis and RSA to navigational affordance RDM separately. I suggest combining those two approaches, using commonality analysis.

9) The authors have behavioral data that allows the construction of navigational affordance RDMs for the different image presentation durations. Interestingly, the images were followed by a dynamics mask, making this a backward masking paradigm that has been associated with blocking recurrent processing. As the authors speculate that the early versus late observed correlations to the original navigational affordance RDM to depend on feedforward versus recurrent processing, these additional behavioral RDMs allow testing predictions of that hypothesis (e.g. short duration RDMs should correlate with neural responses early, not late, etc.).

10) There is an inconsistency across analyses, and with respect to previous results, as to whether experimental effects occur in left OPA, right OPA, or both. The authors speculate in the discussion that this might be due to SNR differences related to single versus multishot EPI sequences used. I do not understand this. How would a difference in SNR affect lateralization. Is it not simply that the power of both the previous and the current study is simply not sufficient to exclude likely false negatives?

Version 1:

Reviewer comments:

Reviewer #1

(Remarks to the Author)

I appreciate the care the authors have put into the revision. They have adequately addressed my concerns.

Reviewer #2

(Remarks to the Author)

Thank you for the detailed responses. The manuscript is much improved.

I am still not sure I understand, or agree, about the lateralization issue. If the trajectories are indeed biased to go more to the right than left (Fig 6A), and if the accuracy of the trajectories is not checked (whether they match Bonner or not is not equivalent to participants accurately labelling possible routes of egress, moreover Bonner data shows similar amount to the left and right, unless I am not understanding the plot), I cannot see how the neural effects aren't just biased by unequal sampling.

Reviewer #3

(Remarks to the Author)

The manuscript has improved considerably. The authors have provided additional reader guidance, acknowledged limitations (e.g. ROI definition styles), and clarified results patterns. They provided additional empirical data with presentation times of 264 and 628ms, which provided relevant context, and led the authors to a more balanced interpretation of their results.

Concerning the previous point 5, I suggest a commonality analysis with 3 variables: MEG at time point 1, MEG at time point 2, and navigational affordance. This requires splitting the MEG data in two to avoid spurious correlations (otherwise if time point 1 = time point 2, the correlation is 1 due to shared signal and noise). I do not insist on this point, but I would encourage the authors to see whether the analysis further supports their claims.

Concerning the previous point 8, I wonder why the authors set the condition of 3 contiguous time points as a statistical criterion. Why not use cluster-level correction, or FDR, or FWE? The 3 time points do not appropriately control for multiple comparisons.

Version 2:

Reviewer comments:

Reviewer #2

(Remarks to the Author)

Thank you for the clarification. I would like to request that the analysis included on the left trajectories is added to the main paper, and the wording changed to "trend" rather than "not significant". Additionally, it would help if rather than just

comparing Bonner with Edinburgh on the leftward tendency, there could be a test of whether left vs right trajectories, in the current dataset, were more frequent (Chi square?) and if the Authors are correct, there should be no significant difference and then my concerns about neural lateralization would be very much assuaged.

Reviewer #3

(Remarks to the Author)

The authors have sufficiently addressed my concerns. I wish they had conducted the additional analysis, but as indicated, I will not insist. More importantly, the frequentist statistics are now on a firmer footing. I have no further comments.

Version 3:

Reviewer comments:

Reviewer #2

(Remarks to the Author)

Thank you for the addition into the text. The manuscript is now fantastic and perhaps it's just me, but I think this new analysis strengthens your original claims.

Responses:

We thank all three reviewers for their positive and insightful comments. We have taken these comments on board and made substantial changes to the manuscript. Below we provide a point-point response and all changes to the revised manuscript are highlighted in blue.

Reviewer 1:

This manuscript uses fMRI/MEG fusion plus a behavioral experiment to examine the neural representation of navigational affordances. The fMRI work replicates the results of Bonner & Epstein (2018), showing OPA involvement in coding navigational affordances. The MEG work shows sustained navigational affordances after 270 ms. The fusion analysis shows V2d/V3d and early (33-66 ms) coding of navigational affordances. A behavioral student presented duration-limited scenes (33.3 ms-132 ms) and asked participants to trace a navigational path. Observers were able to do this with brief presentation times for the most part. This is a timely paper that examines this problem from multiple perspectives. The involvement of V2d/V3d is novel, and I appreciate the authors' explicit replication of previous work. I am not an expert in the fMRI methods, so I will not comment on this aspect of the paper.

Response: We thank the reviewer for the positive appraisal of our work.

While I feel positively towards this paper, I have two major concerns with the behavioral experiment. The first is logical. As described in the paragraph that starts on line 277, the authors' motivation for this experiment is to verify the rapid coding of navigational affordances revealed by the MEG data. This risks conflating presentation duration with precessing time. Presenting a scene for 33.3 ms and finding that an observer can assess the navigational paths is not the same as the navigational paths being coded in 33.3 ms as the MEG shows. I recommend exempling the paper from Van Rullen (2011) on this fallacy. This interpretation also shows up in lines 309-310 ("Taken together, we observed strong evidence that the visual information relevant for navigational affordances can be extracted in the briefest of glances (~33-66ms)" I would also urge the authors to be more careful in this statement, given the lack of statistical robustness in the earliest presentation time.

Response: We are grateful to the reviewer for raising this issue (and for the reference). We now include two additional time-dependent conditions (264ms & 528ms) and a new analysis of the relationship between these time-dependent conditions in the revised manuscript. This new analysis reveals that the representations of navigational affordances appear robust with presentation times of 132ms and longer. We now explicitly discuss how we believe one should interpret the behvaioural and MEG responses taking into account the differences between presentation time and processing time in the revised manuscript.

My second issue with the behavioral experiment is technical. On lines 568-569 the authors note that the display was "HP EliteDesk monitor (resolution: 1920 × 1080 pixels; refresh rate: 60 Hz; visible display size: 531.36 mm × 298.89 mm)." This means that in the fastest condition, images

were on screen for two monitor refreshes. Dropped frames can be very common on PsychoPy (especially on Mac computers, but I have observed it across Linux and Windows machines, too). A dropped frame during a scene presentation would create 50% error and could be a driver of the 33.3 ms condition not showing statistical significance.

As neat as this behavioral paradigm is, the logical and technical issues lead me to believe that this would be a stronger paper without this experiment.

Response: We acknowledge the possibility of dropped frames, which as the reviewer points out, could introduce errors that are larger for the shortest time-dependent condition (33ms or 2 frames per scene). The reviewer is correct that a single dropped frame during the 33ms condition would create a 50% error. This error percentage would decrease as the duration in frames increased (25% at 66ms, 12.5% at 132ms, 6.25% at 264ms and 3.12% at 528ms), assuming only a single frame were dropped. We did not receive reports of dropped frames throughout the experiment. Nevertheless, we have checked for the possibility of dropped frames by running PsychoPy's in-built function timeByFramesEx.py on the same machine that was used to collect the behavioural data. This in-built function runs 500 frames of a simple visual experiment and detects dropped frames should they occur. We did not detect any dropped frames and the mean frame rate was 16.67ms.

Figure R1: Test for dropped frame in PsychoPy. Images are from the output of timeByFramesEx.py. The top image plots the time for each of 500 frames - no frames were

dropped. The bottom left histogram shows the distribution of if frame intervals with the bottom right showing the mean inter flip intervals.

With respect to the non-significant effect at 33ms specifically, we note that when compared to the replication RDM (i.e., unrestricted condition) the correlation with the 33ms RDM falls to the right of the 95th percentile of the null distribution. It is only when compared to the original Bonner RDM where this is not the case. Critically here, the 33ms RDM is the same in both comparisons. The differences in the correlations reflect the differences between the Bonner and replication RDMs, respectively.

We respect the reviewers position here, but believe that after more clearly distinguishing between presentation time and processing time, as well as adding two more time-dependent conditions that the behavioural data is a valuable contribution to the manuscript. Indeed, the inclusion of two longer time-dependent conditions (264ms and 528ms) prompted us to look at the similarity in the representation of navigational affordances across conditions. Here, we computed the dissimilarity (1-Pearson's r) between all pairs of navigational affordance RDMs. The resulting matrix (pasted below for reference and **Fig. 6C**) represents the relationship between each time-dependent condition and reveals several key insights.

First, it is clear that the 33ms RDM is different from all others with the difference increasing with increasing durations. Second, and more important, there is a clear grouping in the lower right quadrant of the matrix for the 132, 264 & 528ms conditions. This grouping of low dissimilarity values suggests that the navigational affordance representations are robust across these timing conditions. This result implies that there is little change to the navigational affordance representations beyond 132ms of stimulus presentation. So, while it is possible to observe significant correlations at shorter stimulus durations, those representations are emerging, but not strong.

The robustness of the representations at 132ms presentation and beyond is also more in-line with our MEG and MEG-fMRI fusion analyses that indicate significant effects emerging ~110ms post stimulus onset. We now discuss these time-dependent results and how they relate to the neural recordings in more detail in the revised manuscript.

Figure R2: Time-dependent RDM-RDM matrix. Cells represent the dissimilarity ($1 - \text{Pearson's } r$) between all pairs of time-dependent navigational affordance RDMs. RDMs that are more similar are shown in blue with RDMs that are more different in yellow. Three clusters are evident. First, the 33ms condition is very different from all others. Second, the 66ms condition is very different from the 33ms condition and moderately different to all others. Third, the 132, 264 & 528ms conditions are all similar to each other suggesting stable representations.

Reviewer 2:

This paper investigates a concept that has not been extensively studied despite it being very relevant for all motile animals and decades old: affordances. The paper does an in depth study on where affordances are processed in the brain, and the Authors present a clear manuscript with the use of cool approaches and analyses. I have a couple questions/comments (in order they came up while reading the manuscript):

1. Would you predict that species that may be more tuned into upper visual field (eg: flying/underwater animals, especially eg: killer whales or other marine hunters who find prey above them) may have different affordances? Different representations in the brain?

Response: This is an intriguing question. Some researchers (e.g., [1]) have speculated on the link between behaviours of certain species (including humans) and dependencies to either the upper or lower visual fields. We are not aware that this has been extended to consider navigational affordances and, whilst an interesting question, believe it is beyond the scope of the current manuscript.

2. Is action without locomotion the same as affordances?

Response: We are unsure if a consensus has been reached regarding the various facets of affordance behaviour. The focus of the current manuscript was on navigational affordances specifically. We, like Bonner et al previously [2], operationalised this as identifying available routes of egress through scenes. Action affordances without locomotion may well be analogous to navigational affordances. If this were the case, one would predict similar effects. We now include a note of this in the revised manuscript.

3. Why left hemisphere effects only? Could this be related to the Figure 5A middle plot of behavioural trajectories which seem to show few turns to the left in your dataset?

Response: The stronger effects in the left hemisphere intrigued us also. Although effects were present in the right hemisphere, they were in general weaker. It is not immediately obvious why the fMRI, and MEG-fMRI fusion analyses would be left hemisphere biased. One possibility is that this reflects differences in the temporal signal to noise (tSNR) between hemispheres. Consistent differences in tSNR could lead to the same pattern of responses in the left and right hemispheres, despite lower overall magnitudes in the right. We now discuss this as a possibility in the revised manuscript.

Importantly, the larger effects observed in the left hemisphere are not likely to reflect the behavioural trajectories as the available routes of egress to the left, middle and right were largely shared across the 50 scenes.

4. Why would LO1/2 effect not be stronger if the argument being made is about lower visual fields' importance in affordance coding?

Response: We recognise the reviewer's point here. Going into this project the prediction was that, if OPA played a crucial role in navigational affordance coding then of its sub-divisions, LO1 and LO2 should show the strongest coding. That prediction was not confirmed by the data. Instead, LO1, LO2, V3A and V3B all showed stronger coding of navigational affordances relative to V7. Whilst LO1 and LO2 (in our data) exhibit a clear lower visual field bias, V3A and V3B also represent the lower visual field, and all (i.e., LO1, LO2, V3A & V3B) do so to a larger degree than V7. So whilst we did not find evidence in favour of a specific advantage for LO1 and LO2, the data do appear consistent with the premise that a lower visual field representation may confer an advantage over an upper visual field one for coding navigational affordances. We now include a more in depth discussion of this pattern of results in the revised manuscript.

5. What is the reasoning behind using all MEG sensors?

Response: Computing effects across all MEG (or EEG) sensors is a common analysis approach when performing MEG-based RSA and/or MEG-fMRI fusion. This is in-part because it is not always clear which sensors one would a-priori expect the effect to manifest, as the mapping between locations within the brain and sensors outside of the brain is non-trivial. We also selected

to analyse all sensors as to be consistent with prior EEG/MEG approaches taken by our group [3] and work that explored the time-course of navigational [4,5] and locomotive affordance coding [6].

6. Why not check significant time periods with cluster based stats (which in my opinion are more common for M/EEG analyses) instead of the Bayesian factor?

Response: A Bayesian framework was selected to be consistent with our prior EEG-RSA and EEG-fMRI fusion analyses [3]. We also recognise that Bayesian analyses are becoming an increasingly common approach to analysing neuroscience data [7] and for time-resolved effects in particular [8–11].

Comparing the results of Bayesian and frequentist approaches can also be difficult. For example, it is not clear to us how one should interpret a result if an effect is present in one framework (e.g., Bayesian) but not another (e.g., frequentist). That said, we recognise that a frequentist approach is often adopted and have reanalysed the MEG-RSA and MEG-fMRI fusion analyses using the following permutation and cluster based approach. We note that the two approaches highlight largely similar time windows of '*significant*' effects.

For each participant and timepoint we randomly shuffled the RDM values before computing the correlation between the shuffled MEG RDM and the navigational affordance RDM (1000 times). These permuted correlation coefficients were then averaged across participants. Next, for each timepoint we assessed whether the observed correlation between the MEG and navigational affordance RDM was greater than the 95th percentile of that timepoint's average permuted null distribution. We considered windows with more than 3 contiguous timepoints (i.e., 3ms) to be significant. The same approach was adopted for the MEG-RSA analysis taking into account the gist model and for the MEG-fMRI fusion and commonality analyses.

Given the potential pitfalls of interpretation mentioned above, we retain our original Bayesian analyses, but also indicate significant timepoints as identified using the approach described above for the interested reader.

7. Please include an image of OPA sub-areas tested (LO1, LO2, V3A, V3B & V7) to compare to Figure 3.

Response: We agree with the reviewer and now include a schematic of OPA and its retinotopic subdivisions in the revised manuscript (**Fig. 1A**). The figure panel is also pasted below for reference.

Figure R3: OPA and retinotopic sub-divisions. A partially-inflated representation of the left hemisphere of a representative participant is shown. The group-average OPA from the scene face localiser (Scenes>Faces, $p < 0.0001$, uncorrected) is overlaid in green. The borders of LO1, LO2, V3A, V3B and V7 taken from a probabilistic atlas [12] are overlaid in white. Consistent with prior work [13,14] on average OPA spatially overlaps these five retinotopic maps to differing degrees.

8. Figure 5b timing in the image is different from what is in text?

Response: Fixed in the revised manuscript. We now also include two more time-dependent conditions (264ms & 528ms) in the behavioral experiment.

9. How was the accuracy of extraction of path trajectories established?

Response: Our experimental procedure for measuring path trajectories followed the approach outlined by Bonner et al. participants were instructed to draw visible paths on the ground without drawing on/up stairs or behind barriers where trajectories could be inferred but were not directly visible. We did not manually correct individual participant path trajectories for accuracy, but note that our replication RDM is strongly correlated with the RDM from Bonner et al, suggesting very similar trajectories were defined.

10. How do the differences in timing in Bonner (2sec) change the comparisons?

Response: The current fMRI experiment applied the same stimulus presentation paradigm as reported for experiment 2 of Bonner et al. In Experiment two, these scene images were presented for 1.5s each (note, that in Experiment 1 of Bonner et al, the artificially created scene images were presented for 2s). We confirm that our stimulus presentation times during fMRI match those of experiment 2 of Bonner et al.

11. Because the earliest time of the effect in MEG is ~100ms and not <66ms, I don't think it's fair to say that the neural effects match the behavioural speed of extraction (and see point 9 above - what does effect at 33ms really mean)?

Response: This point is well taken (we note similar points made by Reviewer's 1 and 3). We now more clearly separate out our discussion of presentation time and processing time in the revised manuscript.

The reviewer asks specifically what the effect we report at 33ms *really* means. Taking the above point in-mind, we interpret this effect as suggesting that the human visual system is capable of extracting the possible routes of egress within a scene (better than chance) with a presentation time of as little as 33ms. That is not to say that the computation takes 33ms - It almost certainly takes much longer than this - but simply that 33ms of presentation is all that is required for the computation to be performed at above chance levels.

The comments raised by all three reviewers prompted us to not only include two additional time-dependent conditions (264ms & 528ms), but also to probe the relationship between those time-dependent conditions. As discussed above in response to Reviewer 1, this analysis reveals that the representations of navigational affordances appear similar (i.e., robust and stable) at 132ms presentation and beyond. In contrast, the 33ms condition is very dissimilar (i.e., emerging but not strong) relative to all others. We now discuss the robustness of these effects and how we believe one should interpret the earlier time-dependent conditions in more detail in the revised manuscript.

12. This is just personal but I don't really like the abbreviation "NA" for navigation affordance coding (Figure 1), it's just a little confusing because NA is commonly "not applicable"

Response: We agree. This has been edited in the revised manuscript.

13. I am not sure I agree the differences found between Bonner and there are from the fMRI preprocessing, but more likely to be related to performance and/or timing (point 10 too). Also just to be clear, when it is stated that the "behavioral RDM from Bonner et al [11]" you are referring to Fig 2 in the paper.

Response: We thank the reviewer for their thoughts here (we also note Reviewer 3 raises a related point). To clarify, the main differences between the current and Bonner results revolve around the searchlight analyses. In Bonner, significant evidence for coding of navigational affordances was only found in the vicinity of OPA in the right hemisphere, whereas in our data,

we observed evidence for navigational affordance coding in both the left and right hemispheres. The discrepancy in the fMRI searchlight analyses is what drove us to speculate as to whether differences in data acquisition protocol could underpin this. We have removed this section in the revised manuscript.

All of the fMRI and MEG analyses that compare neural data to navigational affordances use the navigational affordance RDM taken from Experiment two of Bonner et al (Figure 3C in Bonner et al).

Reviewer 3:

Zamboni et al. conducted a study on the coding of navigational affordances in the human brain. For this they combine fMRI, MEG and behavioral data. Their two main claims are that cortical regions V2d/V3d prior to OPA already code for navigational affordances, and that navigational affordances can be extracted from images even with brief glances lasting as little as 66ms. This is an interesting and thought-provoking study, which adds substantially to the literature on navigational coding. It builds a strong foundation by first replicating previous work before moving on to novel analyses.

I have several questions the answers to which should enable me to better determine the strengths of the empirical evidence. I also have suggestions for further analyses and clarifications that may strengthen the manuscript even further.

1) I am uncertain what the exact claims of the authors are about the relationship between the two major claims they make (see above). The abstract reads "...we demonstrate that navigational affordances can be extracted within 33-66ms, consistent with navigationally relevant signals emerging early in visual processing.". How is that consistent or inconsistent? A strong interpretation would be that being extractable in a brief glance is a counterpart to signals emerging early in visual processing – but are these not unrelated measures? In the discussion instead those two topics seems to be discussed in a fashion not indicating a close connection. So what is the stance of the authors?

Response: We apologise for any confusion. We note also that Reviewer 1 raises a related point with respect to presentation time and processing time.

We have taken these responses on board and collected two further time-dependent datasets (264ms and 528ms). Crucially, we have also looked into the relationship between these time-dependent RDMs. As shown in Figure R1 above (and **Fig. 6C** in the revised manuscript) this analysis reveals that the navigational affordance representations appear robust at presentation times of 132ms and beyond. In contrast, the representations at either 33ms or 66ms appear to be emerging, but not strong. Specifically, the 33ms RDM appears representationally distinct relative to the other conditions.

So, while our initial submission highlighted the significant correlation observed with as little as 33ms presentation time, our revised manuscript provides a more nuanced interpretation of these data. Specifically, with the addition of extra time-dependent conditions and the analyses suggested by the Reviewers, we now highlight that navigational affordance representations appear to be robust with stimulus durations of 132ms and beyond and that the ability to observe significant effects prior to that should be treated with caution. Further, we highlight that the emergence of robust behavioural representations at 132ms aligns well with the MEG-RSA, MEG-fMRI fusion and commonality analyses (described in response to point 8 below).

2) The results section jumps right in with the first analysis without describing the experimental design and task in any detail. The authors should add a paragraph on what actually happened: what was the paradigm, what was the task, what kind of data was measured to guide the reader.

Response: We completely agree and now include an explanatory paragraph outlining the tasks and types of data being measured at the end of the introduction in the revised manuscript.

3) It is confusing that some of the region definitions that are possible based on retinotopic mapping are based on participant-specific pRF estimates, and some are not, instead relying on anatomical atlases. Those atlases can only be less precise than participant-specific mapping. This is important when for example differentiating OPA from more posterior regions. For the claims of the manuscript this is not a major downside as arguably, even with better localization, there would be no other regions posterior to OPA than V2/3d. However, the limitation should be acknowledged and possible implications discussed.

Response: We agree with the reviewer. Our intention initially was to define all OPA subdivisions within each participant and therefore maximise the precision of this analysis. Unfortunately, not all of the retinotopic divisions of OPA were identifiable in all participants and so we decided to adopt a probabilistic approach for that analysis to equate not only the number of included maps, but also the number of nodes. In contrast, V1, V2d and V3d were identifiable in all participants using participant-specific pRF data. We now include a discussion of the implications of these choices in the revised manuscript.

4) l. 134 and before. The emerging results pattern is complex. What do the authors finally conclude from the results? Please clarify.

Response: We are grateful for the opportunity to clarify. If we understand the reviewer correctly, lines 134 and before (in the original submission) referred to the fMRI-RSA results that considered OPA as a whole and then the retinotopic subdivisions of OPA (note Reviewer 2 also raises a similar point).

First, we emphasise that our OPA-RSA analyses produced results that were fully consistent with prior work that considered OPA as a whole. Second, we took the novel approach of considering the role that the individual retinotopic sub-divisions play in coding navigational affordances. Going into this analysis, we predicted that LO1 and LO2 might play a strong role in this process as these

maps reliably show lower visual field biases [13,14], whereas both V3A and V3B show full hemifield representations and V7 an upper visual field bias (Silson et al., 2018; Scrivener et al., 2024). That prediction was not supported by the data. Instead, LO1, LO2, V3A and V3B all showed stronger coding of navigational affordances relative to V7. So, whilst we did not find evidence in favour of a specific advantage for LO1 and LO2, the data do appear consistent with the premise that a lower visual field representation may confer an advantage over an upper visual field one for coding navigational affordances. Indeed, testing the fMRI-RSA correlations against zero (i.e., no correlation) reveals a significant effect in LO1, LO2 V3A and V3B (Bonferroni corrected), but not V7. We now include this analysis in the revised manuscript.

It appears that a lower visual field representation is crucial to the coding of navigational affordances. Had we been able to identify each retinotopic map in each participant, we could have attempted to separate the responses from the upper and lower visual field representations of V3A and V3B, for example. The prediction would be for larger correlations in the lower versus upper visual field portions of both maps. Unfortunately, this is not possible with the use of probabilistic maps. We discuss how one should interpret this result in more detail in the revised manuscript.

5) Assessing the temporal coding of navigational affordances the authors make a point differentiating early from late responses. I suggest a temporal generalization analysis to determine the relationship between the representations at earlier versus later time points, across analyses in the manuscript.

Response: We thank the reviewer for raising this possibility. We note however that to our knowledge temporal-generalisation analyses are almost always conducted in the context of decoding accuracy [15], where one can train and test a classifier at all time points. In contrast, the current analyses ask how the MEG patterns over time relate to a fixed behavioral model using RSA. It is not clear to us how we could perform temporal-generalisation within the RSA framework. One paper [16] whose initial analyses were also time-based RSA did perform temporal-generalisation, but here they simply correlated the sensor activity at each epoch with the sensor activity at every other epoch without reference to the model used in the preceding RSA analysis. Implementing such an analysis indicates when the MEG patterns relate to each other over time, but not how they subsequently relate to the navigational affordance model. If the reviewer knows of an approach that we do not, we would be happy to implement it in further revision.

6) Based on the results reported in Fig. 2D the authors claim that navigational affordances are already coded at the border of V2d/V3d. For one, I am uncertain about whether the expression “at the border” is justified, given that we are dealing with group-average anatomically determined borders. Second, there are also strong effects posterior to V2, in V1, or, using the authors’ terms, at the border of V1 and V2d (especially in the right hemisphere. How do the authors interpret this? I also suggest that the authors add a MEG-fMRI fusion analysis for V1.

Response: We thank the reviewer for raising these points. First, we acknowledge the concern regarding assigning precise labels to group-average anatomically determined borders. We have modified our language in the revised manuscript, now referring to dorsal (i.e., V1d, V2d & V3d)

and ventral (i.e., V1v, V2v & V3v) early visual cortex, respectively. In this regard, we interpret the searchlight analyses as suggestive of stronger navigational affordance coding in dorsal versus ventral early visual cortex without overly interpreting the precise anatomical location of its peak. We now also include MEG-fMRI fusion analyses for V1 in the revised manuscript (**Fig. 4A, 4B**, also pasted below for reference).

Figure R4: MEG-fMRI fusion for left and right V1. A) MEG-fMRI fusion timeseries for left V1. Stimulus onset (time=0ms) and stimulus offset (time=500ms) are shown by vertical lines. **B)** Same as (A) but for right V1. All: timepoints with Bayes Factors that were greater than three are indicated by the coloured circles along the bottom bar, suggesting that the alternative hypothesis of a fusion correlation is at least three times more likely than the null hypothesis ($BF_{10} > 3$). Dark grey circles indicate Bayes evidence in favour of the null hypothesis ($BF_{10} < 1/3$). All: black symbols along the x-axis indicate significant timepoints under a permutation and cluster based frequentist approach.

7) Figure 4 reports strong effects after stimulus offset. How is this to be interpreted?

Response: We are grateful for the opportunity to clarify. Figure 4 depicted the MEG-fMRI fusion timecourses for left and right V2d and V3d, respectively. As the reviewer highlights, this analysis revealed strong effects emerging ~110ms post stimulus onset. We interpret this effect as simply reflecting a strong similarity in the representational structure elicited across the set of 50 scenes in these ROIs (captured by the fMRI RDMs) and at these time-points (captured by the MEG RDMs). We note also, that this analysis was not a-prior planned, but came as a result of the fMRI searchlight analysis that highlighted strong evidence for navigational affordance coding in dorsal early visual areas.

Because the MEG-fMRI fusion analyses themselves are not related to the navigational affordance RDM specifically, we cannot draw strong conclusions about the time course of navigational affordance coding from these data. We are grateful in this regard for the suggestion of the

reviewer to perform commonality analyses in order to relate the MEG-fMRI fusion to the navigational affordance RDM more directly. This analysis, described and plotted in response to point 8 below, highlights unique variance attributable to the navigational affordance model that emerges early following stimulus presentation in V2d, V3d and OPA.

8) The authors conduct fusion analysis and RSA to navigational affordance RDM separately. I suggest combining those two approaches, using commonality analysis.

Response: We are grateful to the reviewer for this suggestion. We implemented a commonality analysis approach based on the details described by Hebart and colleagues [17]. Under this framework, for each ROI, we compared the coefficients of determination (R^2) at each timepoint between 1) the group-averaged MEG and fMRI RDMs and 2) the group-averaged MEG and fMRI RDMs when partialing out the navigational affordance RDM. In order to determine significant time-points we implemented the cluster based analysis described above. For every time-point we randomly shuffled the MEG RDM (1000 times) before calculating the correlation (Spearman's rho). Following the approach outlined previously [17], we computed the difference between these two coefficients of determination (for the observed and permuted data) and for each timepoint we assessed whether the observed correlation was greater than the 95th percentile of that timepoints permuted null distribution. We considered windows with more than 3 contiguous timepoints to be significant. This analysis identifies time-points where the shared variance between MEG and fMRI data can be uniquely attributed to the navigational affordance model. We now include the commonality analyses for V2d, V3d and OPA in the revised manuscript (**Fig. 5**, and pasted below for reference).

Figure R5: Commonality analysis for V2d, V3d and OPA. (A) The mean commonality time-series is plotted for the left hemisphere V2d (green), V3d (blue) and OPA (red). Significant timepoints are identified below colourcoded for each ROI. (B) same as (A) but for the right hemisphere.

9) The authors have behavioral data that allows the construction of navigational affordance RDMs for the different image presentation durations. Interestingly, the images were followed by a

dynamics mask, making this a backward masking paradigm that has been associated with blocking recurrent processing. As the authors speculate that the early versus late observed correlations to the original navigational affordance RDM to depend on feedforward versus recurrent processing, these additional behavioral RDMs allow testing predictions of that hypothesis (e.g. short duration RDMs should correlate with neural responses early, not late, etc.).

Response: We thank the reviewer for this suggestion. We have implemented a version of this analysis in the following approach, noting that this is an additional tweak to the analyses that we had not a-priori selected. Across the OPA, V2d and V3d MEG-fMRI fusion analyses there were broadly two time-windows within which there was evidence for significant fusion: an early time window (~100-450ms post stimulus onset) and a later time window (~800-1000ms post stimulus onset). To explore the idea that the independently derived time-dependent RDMs could capture early and later processes, we first computed the correlation at each timepoint between the MEG RDMs and both the 33ms and 528ms RDMs. That is, our earliest and latest time-dependent RDMs. Next, we calculated the mean correlation within each epoch (early, late) for both RDMs. On average, correlations were positive for the 33ms RDM ($M=0.007$) and negative for the 528ms RDM ($M=-0.004$) in the early time-window, but these were not significantly different ($t(13)=1.06$, $p=0.30$). With the late time-window, correlations were negative for the 33ms RDM ($M=-0.008$), and positive for the 528ms RDM ($M=0.007$), but again these were not significantly different ($t(13)=0.79$, $p=0.43$). Given the lack of significant effects we have selected not to include the analysis below in the revised manuscript.

Figure R6: MEG-RSA with early (33ms) and late (528ms) time-dependent RDMs. A) The mean (plus standard error) MEG-RSA timecourses are plotted for the 33ms RDM (green) and the 528ms RDM (magenta). Two time windows informed by the MEG-fMRI fusion analyses are shown by the vertical gray bars, one early (100-450ms) and one late (800-1000ms). **B)** Bars represent the mean MEG-RSA correlation in each time-window for both the 33ms (green bars) and 528ms

(magenta bars) RDMs. Despite different numerical correlations on average, these were found not to be significantly different in either time-window ($p > 0.05$, in both cases).

10) There is an inconsistency across analyses, and with respect to previous results, as to whether experimental effects occur in left OPA, right OPA, or both. The authors speculate in the discussion that this might be due to SNR differences related to single versus multishot EPI sequences used. I do not understand this. How would a difference in SNR affect lateralization. Is it not simply that the power of both the previous and the current study is simply not sufficient to exclude likely false negatives?

Response: We thank the reviewer raising this (we also note Reviewer 2 makes a similar point). To clarify, the main differences between the current and Bonner results revolve around the searchlight analyses. In Bonner, significant evidence for coding of navigational affordances was only found in the vicinity of OPA in the right hemisphere, whereas in our data, we observed evidence for navigational affordance coding in both the left and right hemispheres. The discrepancy in the fMRI searchlight analyses is what drove us to speculate as to whether differences in data acquisition protocol could underpin this. That is, could it be the case that there was insufficient SNR in the single EPI case to detect effects on the left hemisphere? That said, we accept the reviewer's position and have removed this speculative discussion from the revised manuscript.

References

1. Previc, F.H. (1990) Functional specialization in the lower and upper visual fields in humans: Its ecological origins and neurophysiological implications. *Behav. Brain Sci.* 13, 519–542
2. Bonner, M.F. and Epstein, R.A. (2017) Coding of navigational affordances in the human visual system. *Proc. Natl. Acad. Sci.* 114, 4793–4798
3. Scrivener, C.L. *et al.* (2025) Visual imagery of familiar people and places in category selective cortex. *Neurosci. Conscious.* 2025, niaf006
4. Harel, A. *et al.* (2022) Early Electrophysiological Markers of Navigational Affordances in Scenes. *J. Cogn. Neurosci.* 34, 397–410
5. Dwivedi, K. *et al.* (2024) Visual features are processed before navigational affordances in the human brain. *Sci. Rep.* 14, 5573
6. Bartnik, C.G. *et al.* (2024) Representation of locomotive action affordances in human behavior, brains and deep neural networks *Neuroscience*
7. O'Reilly, J.X. *et al.* (2012) How can a Bayesian approach inform neuroscience? *Eur. J. Neurosci.* 35, 1169–1179
8. Teichmann, L. (2022) An Empirically Driven Guide on Using Bayes Factors for M/EEG Decoding. *Aperture Neuro* 2, 1–10
9. Teichmann, L. *et al.* (2022) The nature of neural object representations during dynamic occlusion. *Cortex* 153, 66–86
10. Moerel, D. *et al.* (2024) Selective attention and decision-making have separable neural bases in space and time. *J. Neurosci.* DOI: 10.1523/JNEUROSCI.0224-24.2024
11. Corriveau, A. *et al.* (2023) Sustained neural representations of personally familiar people and places during cued recall. *Cortex* 158, 71–82
12. Wang, L. *et al.* (2015) Probabilistic Maps of Visual Topography in Human Cortex. *Cereb. Cortex* 25, 3911–3931

13. Silson, E.H. *et al.* (2016) Evaluating the correspondence between face-, scene-, and object-selectivity and retinotopic organization within lateral occipitotemporal cortex. *J. Vis.* 16, 1–21
14. Scrivener, C.L. *et al.* (2024) Retinotopy drives the variation in scene responses across visual field map divisions of the occipital place area. *J. Vis.* 24, 10
15. King, J.-R. and Dehaene, S. (2014) Characterizing the dynamics of mental representations: the temporal generalization method. *Trends Cogn. Sci.* 18, 203–210
16. Hubbard, R.J. and Federmeier, K.D. (2021) Representational Pattern Similarity of Electrical Brain Activity Reveals Rapid and Specific Prediction during Language Comprehension. *Cereb. Cortex* 31, 4300–4313
17. Hebart, M.N. *et al.* (2018) The representational dynamics of task and object processing in humans. *eLife* 7, e32816

Responses to reviewers:

We are grateful again to all three reviewers for their constructive comments. We have taken these comments on board and provide below a point-point response. All edits to the manuscript are highlighted in blue.

Reviewer 1:

I appreciate the care the authors have put into the revision. They have adequately addressed my concerns.

Response: We thank the reviewer for their positive appraisal.

Reviewer 2:

Thank you for the detailed responses. The manuscript is much improved.

Response: We thank the reviewer for the constructive comments.

I am still not sure I understand, or agree, about the lateralization issue. If the trajectories are indeed biased to go more to the right than left (Fig 6A), and if the accuracy of the trajectories is not checked (whether they match Bonner or not is not equivalent to participants accurately labelling possible routes of egress, moreover Bonner data shows similar amount to the left and right, unless I am not understanding the plot), I cannot see how the neural effects aren't just biased by unequal sampling.

Response: We are grateful for the opportunity to clarify, and we apologise for any confusion. The trajectory heat map and angular histogram data plotted in the top row of Figure 6A are for a single scene only. They were included to illustrate the analysis pipeline that was applied to all scenes. The reviewer is correct that for this scene specifically there is a higher frequency of left trajectories (i.e., angles 0-90) in the Bonner data than in the Edinburgh replication data. To test that this is not systematic across the 50 tested scenes, we have calculated the mean leftward frequency for all 50 scenes in the Bonner and Edinburgh replication datasets. There was no significant difference in mean leftward frequency across scenes (\$t(49)=2.00\$, \$p=0.056\$ ). This result suggest that unequal sampling of path trajectories is unlikely to explain the differences in lateralisation of the fMRI searchlight data between the two studies.

Reviewer #3 (Remarks to the Author):

The manuscript has improved considerably. The authors have provided additional reader guidance, acknowledged limitations (e.g. ROI definition styles), and clarified results patterns. They provided additional empirical data with presentation times of 264 and 628ms, which provided relevant context, and led the authors to a more balanced interpretation of their results.

Response: We thank the reviewer for their positive feedback.

Concerning the previous point 5, I suggest a commonality analysis with 3 variables: MEG at time point 1, MEG at time point 2, and navigational affordance. This requires splitting the MEG data in two to avoid spurious correlations (otherwise if time point 1 = time point 2, the correlation is 1 due to shared signal and noise). I do not insist on this point, but I would encourage the authors to see whether the analysis further supports their claims.

Response: We thank the reviewer for the clarification. However, as this did not form part of our a-priori analysis framework we believe exploring this further is beyond the scope of the current manuscript.

Concerning the previous point 8, I wonder why the authors set the condition of 3 contiguous time points as a statistical criterion. Why not use cluster-level correction, or FDR, or FWE? The 3 time points do not appropriately control for multiple comparisons.

Response: We are grateful for the opportunity to expand here. We accept that our prior implementation did not appropriately control for multiple comparisons and have re-calculated the frequentist statistics using the following approach. Correction for multiple comparisons was implemented in CoSMoMvPA (cosmo motecarlo cluster stat, Oosterhof et al., 2016) using threshold-free cluster-enhancement (Smith & Nichols, 2009; Menon & Khatami, 2013; Maris et al., 2007). For each analysis we ran 10,000 permutations. For a one-sample t-test and an alpha of 0.05, z-score values greater than 1.65 indicate that the statistic is significant. As we allowed for clustering over time, we can make inferences at the cluster level rather than at individual timepoints. A similar approach was adopted for the MEG-RSA analysis taking into account the gist model, for the MEG-fMRI fusion analyses and the commonality analysis.

In general, we observed similar timepoints from both approaches, yet some differences persisted. For example, in the MEG-RSA analysis comparing the pattern of MEG data with the navigational affordance model, our Bayesian analysis indicated moderate to strong evidence for navigational affordance coding during two distinct time-windows (Figure 2A), but these same time-windows did not survive corrections with the frequentists approach. We note, that our original analyses were planned and implemented under the Bayesian framework for MEG/EEG analyses, and reiterate that making comparisons between analysis frameworks (Bayesian versus frequentists) is non-trivial. As such, we do not overly interpret any differences between the Bayesian and Frequentists approaches but leave both analyses within the revised manuscript for the interested reader.

Responses to reviewers:

We are grateful again to all two reviewers for their constructive comments. Below we provide a point-point response. All edits to the manuscript are highlighted in blue.

Reviewer 2:

Thank you for the clarification. I would like to request that the analysis included on the left trajectories is added to the main paper, and the wording changed to "trend" rather than "not significant". Additionally, it would help if rather than just comparing Bonner with Edinburgh on the leftward tendency, there could be a test of whether left vs right trajectories, in the current dataset, were more frequent (Chi square?) and if the Authors are correct, there should be no significant difference and then my concerns about neural lateralization would be very much assuaged.

Response: We thank the reviewer for the constructive comments. We now include the analysis on path trajectories in the revised manuscript. With respect to comparing the left versus right trajectories in the Edinburgh replication dataset we have taken the following approach. First, we calculated how many of the 50 scenes contained trajectories in only one half of the image (i.e., either all left or all right). None of the 50 scenes met this criterion. This indicates that in all 50 scenes path trajectories were in both left and right halves of the image. Given this, we then calculated the mean angular frequency in the left half or right halves of each scene. Such an analysis provides a means to test for a leftward or rightward bias in path trajectories. A paired t -test indicated the difference in mean angular frequency was statistically non-significant between left and right halves of the scenes ($t(49)=1.10$, $p=0.27$). We now include this and the previous analysis in the revised manuscript.

Lines 516-528: First, we have calculated the mean leftward frequency for all 50 scenes in the Bonner and Edinburgh replication datasets. The difference in mean leftward angular frequency across scenes was statistically non-significant ($t(49)=2.00$, $p=0.056$). This result suggest that unequal sampling of path trajectories is unlikely to explain the differences in lateralisation of the fMRI searchlight data between the two studies. Next, we compared the left versus right trajectories in the Edinburgh replication dataset to rule out a potential trajectory bias in this dataset alone. We calculated how many of the 50 scenes contained trajectories in only one half of the image (i.e., either all left or all right). None of the 50 scenes met this criterion. This indicates that in all 50 scenes path trajectories were in both left and right halves of the image. Given this, we next calculated the mean angular frequency in the left half or right halves of each scene. Such an analysis provides a means to test for a leftward or rightward bias in path trajectories. A paired t -test indicated no significant difference in mean angular frequency between left and right halves of the scenes ($t(49)=1.10$, $p=0.27$).

Reviewer 3:

The authors have sufficiently addressed my concerns. I wish they had conducted the additional analysis, but as indicated, I will not insist. More importantly, the frequentist statistics are now on a firmer footing. I have no further comments.

Response: We thank the reviewer for the constructive comments.